# Theoretical Guarantees on the Best-of-n Alignment Policy

**Ahmad Beirami**[1]  **Alekh Agarwal**[2]  **Jonathan Berant**[1]  **Alexander D'Amour**[1]  **Jacob Eisenstein**[1]  **Chirag Nagpal**[2]
**Ananda Theertha Suresh**[2]

## Abstract

A simple and effective method for the inference-time alignment and scaling test-time compute of generative models is best-of-$n$ sampling, where $n$ samples are drawn from a reference policy, ranked based on a reward function, and the highest ranking one is selected. A commonly used analytical expression in the literature claims that the KL divergence between the best-of-$n$ policy and the reference policy is equal to $\log(n) - (n-1)/n$. We disprove the validity of this claim, and show that it is an upper bound on the actual KL divergence. We also explore the tightness of this upper bound in different regimes, and propose a new estimator for the KL divergence and empirically show that it provides a tight approximation. We also show that the win rate of the best-of-$n$ policy against the reference policy is upper bounded by $n/(n+1)$ and derive bounds on the tightness of this characterization. We conclude with analyzing the tradeoffs between win rate and KL divergence of the best-of-$n$ alignment policy, which demonstrate that very good tradeoffs are achievable with $n < 1000$.

## 1. Introduction

Generative language models have shown to be effective general purpose tools to solve various problems. While many problems can be solved in a zero-shot manner, the output from the so-called *reference* model may not be outright desirable, e.g., it may violate safety rules or may not solve a math problem correctly. *Alignment* (Christiano et al., 2017; Stiennon et al., 2020; Ouyang et al., 2022; Bai et al., 2022) and *test-time compute scaling* (Brown et al., 2024; Snell et al., 2024) aim at remedying this issue by further nudging the outcome to improve a reward function while not drifting

---

[1]Google DeepMind [2]Google Research. Correspondence to: Ahmad Beirami <ahmad.beirami@gmail.com>, Ananda Theertha Suresh <theertha@google.com>.

*Proceedings of the 42$^{nd}$ International Conference on Machine Learning*, Vancouver, Canada. PMLR 267, 2025. Copyright 2025 by the author(s).

too far from the reference model.

Recently, there has been a proliferation of methods for alignment, which include KL-regularized reinforcement learning (Christiano et al., 2017; Ouyang et al., 2022), controlled decoding (Yang & Klein, 2021; Mudgal et al., 2024), SLiC (Zhao et al., 2022), direct preference optimization (Rafailov et al., 2023), and best-of-$n$ finetuning (Touvron et al., 2023). At their core, these methods try to solve the following regularized optimization problem:[1]

$$\max_{\pi(\cdot|\boldsymbol{x})} E_{\boldsymbol{y} \sim \pi(\cdot|\boldsymbol{x})} r(\boldsymbol{x}, \boldsymbol{y}) - \beta D_{\mathrm{KL}}(\pi(\cdot|\boldsymbol{x}) \| \pi_{\mathrm{ref}}(\cdot|\boldsymbol{x})), \quad (1)$$

where $\pi_{\mathrm{ref}}$ denotes a reference language model; $r(\boldsymbol{x}, \boldsymbol{y}) \in \mathbb{R}$ represent a scalar reward associated with response $\boldsymbol{y}$ for prompt $\boldsymbol{x}$; and the KL divergence $D_{\mathrm{KL}}(q(\cdot|\boldsymbol{x}) \| p(\cdot|\boldsymbol{x}))$ is defined as

$$D_{\mathrm{KL}}(q(\cdot|\boldsymbol{x}) \| p(\cdot|\boldsymbol{x})) := E_{\boldsymbol{y} \sim q(\cdot|\boldsymbol{x})} \log \frac{q(\boldsymbol{y}|\boldsymbol{x})}{p(\boldsymbol{y}|\boldsymbol{x})}.$$

Note that Equation (1) has a closed-form solution (Korbak et al., 2022b;a):

$$\pi_{\beta}^*(\boldsymbol{y}|\boldsymbol{x}) \propto \pi_{\mathrm{ref}}(\boldsymbol{y}|\boldsymbol{x}) e^{\frac{1}{\beta} r(\boldsymbol{x}, \boldsymbol{y})}, \quad (2)$$

which defines an exponential family of distributions with nice properties (Yang et al., 2024a). We also define the KL divergence averaged over prompts as $D_{\mathrm{KL}}^{\mu}(q \| p) := E_{\boldsymbol{x} \sim \mu} D_{\mathrm{KL}}(q(\cdot|\boldsymbol{x}) \| p(\cdot|\boldsymbol{x}))$, where $\mu$ is a distribution over prompts. Notice that a small KL divergence between the aligned policy and the reference policy is desired because it implies that the capabilities of the reference policy are largely preserved (Gao et al., 2023; Coste et al., 2024; Eisenstein et al., 2024), which is also theoretically analyzed by Balashankar et al. (2025, Appendix B).

To compare different alignment techniques, it is customary to produce tradeoff curves that measure expected reward (or win rate) as a function of $D_{\mathrm{KL}}(\pi \| \pi_{\mathrm{ref}})$ for some aligned policy $\pi$. Guarantees on the KL divergence capture the preservation of the core capabilities of the model and tighter estimates on the KL divergence help give guarantees that

---

[1]While theoretically we analyze this optimization problem as a function of the prompt $\boldsymbol{x}$, in practice we can only solve it by taking another expectation over a set of prompts $\boldsymbol{x} \sim \mu$.

the model doesn't lose core capabilities that were present in the reference checkpoint. Thus, it is desirable to improve the reward with the least drift measured in KL divergence.

Despite all the advancements in alignment, a simple, popular, and well-performing method for alignment remains to be the *best-of-n* policy (Nakano et al., 2021; Stiennon et al., 2020). In fact, Gao et al. (2023); Mudgal et al. (2024); Eisenstein et al. (2024) show that best-of-$n$ consistently achieves compelling win rate vs KL tradeoff curves, that even dominate those of KL-regularized reinforcement learning and other more involved alignment policies. Llama 2 (Touvron et al., 2023) uses best-of-$n$ as a teacher outcomes to further finetune the base model. Mudgal et al. (2024) extended best-of-$n$ through $q$-learning to block-wise best-of-$n$ decoding. This has also led to recent research on distilling best-of-$n$ into new models (Gui et al., 2024; Amini et al., 2025; Sessa et al., 2025; Qiu et al., 2024). Hughes et al. (2024); Beetham et al. (2024) use best-of-$n$ as an effective method for jailbreaking. Best-of-$n$ is also used as a strong baseline in scaling inference-time compute (Brown et al., 2024; Snell et al., 2024). This overwhelming empirical success motivates our theoretical investigation of the best-of-$n$ alignment policy.

Subsequent to this work, Yang et al. (2024a) provided theoretical reasoning for the performance of best-of-$n$ by showing it achieves asymptotically optimal reward-KL tradeoffs. Gui et al. (2024) characterized the win rate vs KL gap to be small in the asymptotic regime of a language model whose outcomes have infinitesimally small likelihood. Sun et al. (2024) made best-of-$n$ faster through speculative rejection. Mroueh (2024) provided information-theoretic bounds on reward vs KL tradeoffs for best-of-$n$.

**Best-of-$n$.** Let $\boldsymbol{x}$ be a given input prompt to the language model. Let $\boldsymbol{y}_1, \ldots, \boldsymbol{y}_n$ be $n$ i.i.d. samples drawn from $\pi_{\text{ref}}(\cdot|\boldsymbol{x})$. The best-of-$n$ strategy selects[2]

$$\boldsymbol{y} = \boldsymbol{y}_{k^*} \qquad \text{where} \qquad k^* := \arg\max_{k \in [n]} r(\boldsymbol{x}, \boldsymbol{y}_k). \quad (3)$$

This process inherently leads to sampling from a new policy that is aligned to the reward, denoted by $\pi^{(n)}$. Notice that $\pi^{(1)} = \pi_{\text{ref}}$, and increasing $n$ increases the reward at the cost of drifting away from the base model.

Our goal in this paper is to better understand the best-of-$n$ alignmnet policy. In particular, we are interested in theoretical guarantees on $D_{\text{KL}}^{\mu}(\pi^{(n)} \| \pi_{\text{ref}})$ for different values of $n$. A commonly used expression in the literature (Stiennon et al., 2020; Hilton & Gao, 2022; Coste et al., 2024; Gao et al., 2023; Go et al., 2023; Scheurer et al., 2023) claims

$$D_{\text{KL}}^{\mu}(\pi^{(n)} \| \pi_{\text{ref}}) \overset{\text{claim}}{=\!=\!=} \widetilde{\text{KL}}_n := \log(n) - (n-1)/n. \quad (4)$$

This formula is commonly used to demonstrate reward-KL

---

[2]We define $[n] := \{1, \ldots, n\}$.

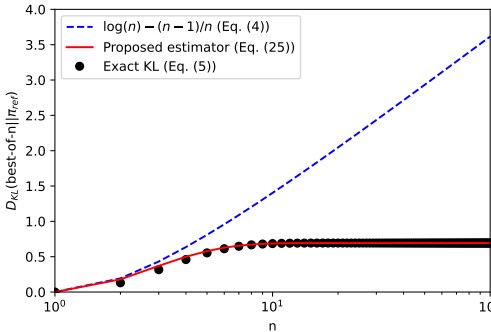

*Figure 1.* The analytical formula $(\log(n) - (n-1)/n)$ (Equation (4)), the exact KL divergence (Equation (5)), and the proposed estimator (Equation (25)), for Example 1, illustrating a case where the gap between the analytical formula and the exact KL divergence is unbounded.

tradeoffs for the best-of-$n$ policy. Let us further inspect this formula using a toy example.

**Example 1.** Consider an unprompted model with $\boldsymbol{x} = \emptyset$ (no input) and binary output, $\boldsymbol{y} \in \{0, 1\}$. Let the two outcomes be equiprobable, i.e., $\pi_{\text{ref}}(0) = \pi_{\text{ref}}(1) = \frac{1}{2}$. Further, let $r(0) = 0$, and $r(1) = 1$, i.e., outcome 1 is more desirable than outcome 0. In this example, we can compute $\pi^{(n)}$ in closed form. Specifically, we can see that $\pi^{(n)}(0) = \frac{1}{2^n}$ and $\pi^{(n)}(1) = 1 - \frac{1}{2^n}$. Thus,

$$D_{\text{KL}}(\pi^{(n)} \| \pi_{\text{ref}}) = \log(2) - h\left(\frac{1}{2^n}\right), \quad (5)$$

where $h(\cdot)$ is the binary entropy function.[3] We compare the exact closed-form expression for KL divergence with the analytical formula in Equation (4). As can be seen in Figure 1 (and is evident from Equation (5)), the true KL is upper bounded by $\log(2)$ for all $n$, whereas $\widetilde{\text{KL}}_n$ grows unbounded as $n \to \infty$. We also report a new estimator for KL divergence that closely mirrors the true KL divergence.

As we learnt from Example 1, the KL divergence between the best-of-$n$ policy and the reference policy may be quite different from what the analytical formula used in the literature suggests. In the rest of this paper, we shed some light on this formula, derive bounds on the KL divergence, and propose a new estimator for the KL divergence that better captures the behavior of the KL divergence. We also theoretically reason about the win rate vs KL tradeoffs for the best-of-$n$ policy, and justify its widespread use in language model alignment.

## 2. Derivation of the Best-of-n Policy

Our first step is to provide a derivation for the best-of-$n$ policy under two simplifying assumptions. Let $r(\boldsymbol{x}, \boldsymbol{y}) \in \mathbb{R}$ represent the scalar reward of response $\boldsymbol{y}$ in context $\boldsymbol{x}$.

---

[3]$h(x) := -x\log(x) - (1-x)\log(1-x)$, for all $x \in (0, 1)$, and $h(0) = h(1) := 0$. Further, note that all logarithms in this paper are to the base $e$.

**Assumption 2.1.** We assume that the reward $r(\boldsymbol{x}, \boldsymbol{y})$ is unique for all $\boldsymbol{x}, \boldsymbol{y}$.

**Assumption 2.2.** Let $\mathcal{Y}^* := \{\boldsymbol{y} \mid \max_{\boldsymbol{x} \in \mathcal{X}} \pi_{\text{ref}}(\boldsymbol{y}|\boldsymbol{x}) > 0\}$. We assume that the language model is such that $|\mathcal{Y}^*| < \infty$, i.e., there are finite possible outcomes (in each context).

Note that Assumptions 2.1-2.2 are fairly non-restrictive and make the presentation of the results clearer.

The following result gives the probability mass function (PMF) of the best-of-$n$ policy.

**Lemma 2.3.** *Under Assumptions 2.1-2.2, for all $n \in \mathbb{N}$, the PMF of the best-of-n policy is given by*

$$\pi^{(n)}(\boldsymbol{y}|\boldsymbol{x}) = \mathcal{F}_{\pi_{ref}}(\boldsymbol{y}|\boldsymbol{x})^n - \mathcal{F}_{\pi_{ref}}^-(\boldsymbol{y}|\boldsymbol{x})^n, \qquad (6)$$

*where for any distribution $\pi$,*

$$\mathcal{F}_\pi(\boldsymbol{y}|\boldsymbol{x}) := P_{\boldsymbol{z} \sim \pi(\cdot|\boldsymbol{x})}[r(\boldsymbol{x}, \boldsymbol{z}) \leq r(\boldsymbol{x}, \boldsymbol{y})], \qquad (7)$$

$$\mathcal{F}_\pi^-(\boldsymbol{y}|\boldsymbol{x}) := P_{\boldsymbol{z} \sim \pi(\cdot|\boldsymbol{x})}[r(\boldsymbol{x}, \boldsymbol{z}) < r(\boldsymbol{x}, \boldsymbol{y})]. \qquad (8)$$

*Proof.* Let $\mathcal{Y}_{\boldsymbol{x}}$ be the set of all possible outcomes of the language model, given prompt $\boldsymbol{x}$, i.e., $\mathcal{Y}_{\boldsymbol{x}} := \{\boldsymbol{y} \mid \pi_{\text{ref}}(\boldsymbol{y}|\boldsymbol{x}) > 0\}$. Further, let $L_{\boldsymbol{x}} := |\mathcal{Y}_{\boldsymbol{x}}| < \infty$ (Assumption 2.2). We order all possible $L_{\boldsymbol{x}}$ outcomes as $\{\widetilde{\boldsymbol{y}}_i\}_{i \in [L_{\boldsymbol{x}}]}$ such that if $r(\boldsymbol{x}, \widetilde{\boldsymbol{y}}_j) > r(\boldsymbol{x}, \widetilde{\boldsymbol{y}}_i)$, then $j > i$. In other words, $\widetilde{\boldsymbol{y}}_1$ is the least desirable outcome associated with the lowest reward, and $\widetilde{\boldsymbol{y}}_{L_{\boldsymbol{x}}}$ is the most desirable outcome associated with the highest reward.

First notice that sampling from $\pi_{\text{ref}}$ is equivalent to sampling $u \sim \mathcal{U}[0, 1]$, and returning $\widetilde{\boldsymbol{y}}_i$, such that

$$\mathcal{F}_{\pi_{\text{ref}}}(\widetilde{\boldsymbol{y}}_{i-1}|\boldsymbol{x}) \leq u < \mathcal{F}_{\pi_{\text{ref}}}(\widetilde{\boldsymbol{y}}_i|\boldsymbol{x}). \qquad (9)$$

Similarly, sampling from the best-of-$n$ strategy is akin to sampling $u_1, \ldots, u_n \overset{\text{i.i.d.}}{\sim} \mathcal{U}[0, 1]$, and returning $\widetilde{\boldsymbol{y}}_i$, such that

$$\mathcal{F}_{\pi_{\text{ref}}}(\widetilde{\boldsymbol{y}}_{i-1}|\boldsymbol{x}) \leq \max_{k \in [n]} u_k < \mathcal{F}_{\pi_{\text{ref}}}(\widetilde{\boldsymbol{y}}_i|\boldsymbol{x}). \qquad (10)$$

On the other hand, we know that the CDF of the maximum of $u_1, \ldots, u_n \overset{\text{i.i.d.}}{\sim} \mathcal{U}[0, 1]$, for all $\tau \in [0, 1]$ is given by

$$P\left[\max_{k \in [n]} u_k \leq \tau\right] = \tau^n. \qquad (11)$$

Hence, for all $n \in \mathbb{N}$, the PMF of the best-of-$n$ policy, denoted as $\pi^{(n)}$ is given by

$$\pi^{(n)}(\widetilde{\boldsymbol{y}}_i|\boldsymbol{x}) = \mathcal{F}_{\pi_{\text{ref}}}(\widetilde{\boldsymbol{y}}_i|\boldsymbol{x})^n - \mathcal{F}_{\pi_{\text{ref}}}(\widetilde{\boldsymbol{y}}_{i-1}|\boldsymbol{x})^n \qquad \forall i \in [L_{\boldsymbol{x}}], \qquad (12)$$

where $\mathcal{F}_{\pi_{\text{ref}}}(\widetilde{\boldsymbol{y}}_0|\boldsymbol{x}) := 0$, and

$$\mathcal{F}_{\pi_{\text{ref}}}(\widetilde{\boldsymbol{y}}_i|\boldsymbol{x}) = \sum_{l \in [i]} \pi_{\text{ref}}(\widetilde{\boldsymbol{y}}_l|\boldsymbol{x}). \qquad (13)$$

The proof is completed by noticing that $\mathcal{F}_{\pi_{\text{ref}}}(\widetilde{\boldsymbol{y}}_{i-1}|\boldsymbol{x}) = \mathcal{F}_{\pi_{\text{ref}}}^-(\widetilde{\boldsymbol{y}}_i|\boldsymbol{x})$. $\square$

Notice that if $n = 1$, then $\pi^{(1)}(\boldsymbol{y}|\boldsymbol{x}) = \pi_{\text{ref}}(\boldsymbol{y}|\boldsymbol{x})$. For any $n$, Lemma 2.3 gives a closed-form expression for $\pi^{(n)}(\boldsymbol{y}|\boldsymbol{x})$, which we will use subsequently to derive theoretical guarantees on the KL divergence and win rate of best-of-$n$.

We also remark that we can extend this PMF to $\pi^{(\tau)}$ for real $\tau \geq 1$. While it may not be immediately clear how to sample from this extension, it is used for best-of-$n$ distillation (Gui et al., 2024; Amini et al., 2025; Sessa et al., 2025) and we also use it to give bounds in Section 7.

# 3. Relations Between the KL Divergence and the Analytical Formula

Our first result shows that the analytical formula is an upper bound on the (context-dependent) KL divergence. The proofs for this result and several subsequent results are relegated to Appendix A.

**Theorem 3.1.** *For any $n \in \mathbb{N}$, and any $\boldsymbol{x}$, let $\widetilde{KL}_n$ be defined in (4). Then,*

$$D_{KL}(\pi^{(n)}(\cdot|\boldsymbol{x})\|\pi_{ref}(\cdot|\boldsymbol{x})) \leq \widetilde{KL}_n = \log(n) - \frac{n-1}{n}.$$

**Corollary 3.2.** *For any $n$, and any prompt distribution $\mu$,*

$$D_{KL}^\mu(\pi^{(n)}\|\pi_{ref}) = E_{\boldsymbol{x} \sim \mu} D_{KL}(\pi^{(n)}(\cdot|\boldsymbol{x})\|\pi_{ref}(\cdot|\boldsymbol{x})) \leq \widetilde{KL}_n.$$

*Proof.* This directly follows from Theorem 3.1. $\square$

Subsequent to this work, Mroueh (2024) has extended this result to a larger class of stochastic processes (with potentially continuous support such as diffusion models) through the application of the strong data processing inequality. In Appendix B, we also extend this result to derive bounds on the KL divergence of the blockwise best-of-$n$ decoding (Mudgal et al., 2024), which generally allows to reach similar reward vs KL tradeoffs with 10x smaller $n$. In the rest of this section, we characterize the gap defined as follows:

$$G_{\text{KL}}^{(n)}(\boldsymbol{x}) := \widetilde{\text{KL}}_n - D_{\text{KL}}(\pi^{(n)}(\cdot|\boldsymbol{x})\|\pi_{\text{ref}}(\cdot|\boldsymbol{x})) \geq 0. \quad (14)$$

## 3.1. Upper Bounds on the Gap

We need a definition to state the upper bound results.

**Definition 3.3.** A model $\pi_{\text{ref}}$ is called $\delta$-bound if $\pi_{\text{ref}}(\boldsymbol{y}|\boldsymbol{x}) \leq \delta$ for all $\boldsymbol{y} \in \mathcal{Y}^*$ and $\boldsymbol{x}$.

In particular, we are interested in characterizing the gap for a $\delta$-bound model for a small $\delta$, which is a model with all outcomes having small likelihoods. Next, we state our main upper bound.

**Theorem 3.4.** *The gap in Equation* (14) *is upper bounded by*

$$G_{KL}^{(n)}(\boldsymbol{x}) \leq 2n(n-1)e^{-H_2(\pi_{ref}|\boldsymbol{x})}, \qquad (15)$$

*where $H_2(\pi_{ref}|\boldsymbol{x})$ is the conditional Rényi entropy of order 2 of the language model given context $\boldsymbol{x}$, and $H_\alpha(\pi)$ for any distribution $\pi$ is defined as*

$$H_\alpha(\pi|\boldsymbol{x}) := \frac{1}{1-\alpha} \log \left( \sum_{y \in \mathcal{Y}^*} (\pi(\boldsymbol{y}|\boldsymbol{x}))^\alpha \right). \qquad (16)$$

**Corollary 3.5.** *Let $\pi_{ref}(\boldsymbol{y}|\boldsymbol{x}) \leq \delta$ for all $\boldsymbol{y} \in \mathcal{Y}^*$, i.e., $\pi_{ref}$ is $\delta$-bound. Then, the gap in Equation* (14) *is upper bounded by*

$$G_{KL}^{(n)}(\boldsymbol{x}) \leq 2n(n-1)\delta. \qquad (17)$$

*Proof.* The proof follows by noticing that $H_2(\pi_{\text{ref}}|\boldsymbol{x}) \geq \log(1/\delta)$ and invoking Theorem 3.4. $\qquad \square$

Intuitively, if the model outcomes are fairly low probability, making it unlikely to get the same sample twice or more in the $n$ outcomes for best-of-$n$, the analytical formula $\widetilde{\text{KL}}_n$ in (4) could be relatively accurate, and the gap is bounded above. In other words, if $\pi_{\text{ref}}$ is a $\delta$-bound model, and $n$ is sufficiently small such that $n^2\delta \ll 1$, then

$$D_{\text{KL}}(\pi^{(n)}(\cdot|\boldsymbol{x})\|\pi_{\text{ref}}(\cdot|\boldsymbol{x})) \approx \log(n) - \frac{n-1}{n}. \qquad (18)$$

This assumption is left implicit in the derivation of Hilton & Gao (2022) for the KL divergence of best-of-$n$.

### 3.2. Lower Bounds on the Gap

In this section, we characterize cases where the gap may be large. To this end, let us define

$$\varepsilon_n := \pi_{\text{ref}}(\boldsymbol{y}|\boldsymbol{x}), \quad \text{where} \quad \boldsymbol{y} \sim \pi^{(n)}(\cdot|\boldsymbol{x}). \qquad (19)$$

Note that $\varepsilon_n$ is a random function of $\boldsymbol{x}$. In the limit as $n \to \infty$, we define

$$\varepsilon_\infty := \pi_{\text{ref}}(\boldsymbol{y}_{\max}(\boldsymbol{x})|\boldsymbol{x}), \qquad (20)$$

where $\boldsymbol{y}_{\max}(\boldsymbol{x}) = \arg\max_{\boldsymbol{y} \in \mathcal{Y}^*} r(\boldsymbol{x}, \boldsymbol{y})$. Notice that $\varepsilon_\infty$ is a deterministic function of $\boldsymbol{x}$.

**Theorem 3.6.** *Let $\varepsilon_\infty > 0$ be defined in Equation* (20). *For $n \in \mathbb{N}$, the gap between the analytical formula in Equation* (4) *and KL divergence is lower bounded by*

$$G_{KL}^{(n)}(\boldsymbol{x}) \geq (1 - (1-\varepsilon_\infty)^n)\Big( \log \frac{n\varepsilon_\infty}{1 - (1-\varepsilon_\infty)^n} - \frac{n-1}{n} \Big)$$
$$- (n-1)(1-\varepsilon_\infty)^n \log(1-\varepsilon_\infty) > 0. \qquad (21)$$

**Corollary 3.7.** *As $n \to \infty$, the gap is lower bounded by*

$$G_{KL}^{(n)}(\boldsymbol{x}) \geq \log(n\varepsilon_\infty) + o_n(\log n). \qquad (22)$$

In particular, when $n\varepsilon_\infty \gg 1$, then the gap grows unbounded as we already observed in Example 1.

## 4. Proposed Estimator for KL Divergence

Motivated by the derivation of the best-of-$n$ policy in Lemma 2.3, we propose a new estimator for the KL divergence. As a warm-up, first notice the following upper bound:

**Lemma 4.1.** *For any $n \in \mathbb{N}$ and any $\boldsymbol{x}$,*

$$D_{KL}(\pi^{(n)}(\cdot|\boldsymbol{x})\|\pi_{ref}(\cdot|\boldsymbol{x}))$$
$$\leq E_{\boldsymbol{y} \sim \pi^{(n)}} \left[ \log \left( \frac{1 - (1 - \pi_{ref}(\boldsymbol{y}|\boldsymbol{x}))^n}{\pi_{ref}(\boldsymbol{y}|\boldsymbol{x})} \right) \right].$$

Therefore we may suggest to use the following *alternate estimator* for KL divergence

$$\widehat{D_{\text{KL,loose}}}(\varepsilon_n) := \log \left( \frac{1 - (1-\varepsilon_n)^n}{\varepsilon_n} \right), \qquad (23)$$

where $\varepsilon_n$ is defined in Equation (19). Note that the expected value of $\widehat{D_{\text{KL,loose}}}(\varepsilon_n)$ is an upper bound on the KL divergence between the best-of-$n$ policy and the reference policy. However, this estimator is loose by an additive constant of $(n-1)/n$, especially when $n\varepsilon_n \ll 1$.

Here we propose a different estimator to close this gap. To derive the estimator, first notice the following result.

**Corollary 4.2.** *Let*

$$d_n(\varepsilon) := (1-\varepsilon)^n \Big( \log n + (n-1)\log(1-\varepsilon) - \frac{n-1}{n} \Big)$$
$$+ (1 - (1-\varepsilon)^n) \log \left( \frac{1 - (1-\varepsilon)^n}{\varepsilon} \right). \qquad (24)$$

*Recall the definition of $\varepsilon_\infty$ in Equation* (19). *Then,*

$$D_{KL}(\pi^{(n)}(\cdot|\boldsymbol{x})\|\pi_{ref}(\cdot|\boldsymbol{x})) \leq d_n(\varepsilon_\infty).$$

Note that Corollary 4.2 could not be directly used to derive an estimator for the KL divergence because we do not observe $\varepsilon_\infty$ when performing the best-of-$n$ policy. Inspired by this result, and given that we can only observe $\varepsilon_n$, we put forth the following practical estimator on the KL divergence.

**Definition 4.3.** *Let $\varepsilon_n$ be defined in* (19). *Then, we propose the following estimator for the KL divergence of the best-of-$n$ policy and the reference policy:*

$$\widehat{D_{\text{KL}}}(\varepsilon_n) := d_n(\varepsilon_n). \qquad (25)$$

Note that the estimator proposed in Definition 4.3 is a random variable that depends on $\varepsilon_n$. We conjecture that in expectation it provides an upper bound on the true KL divergence.

**Conjecture 4.4.** *Let $\varepsilon_n$ be defined in* (19). *Then,*

$$D_{KL}(\pi^{(n)}(\cdot|\boldsymbol{x})\|\pi_{ref}(\cdot|\boldsymbol{x})) \leq E_{\boldsymbol{y} \sim \pi^{(n)}(\cdot|\boldsymbol{x})} \left[ \widehat{D_{KL}}(\varepsilon_n) \right].$$

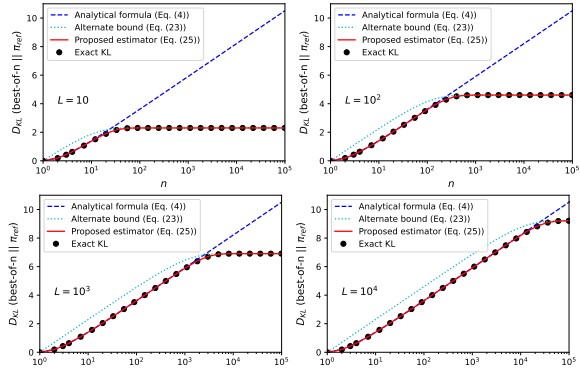

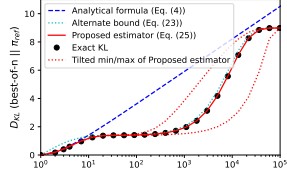

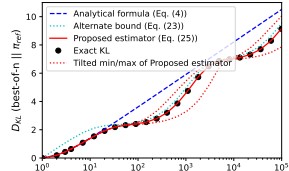

*Figure 2.* The analytical formula $(\log(n) - (n-1)/n)$, Equation (4), the alternate bound, Equation (23), the proposed estimator, Equation (25), and the exact KL divergence, for uniform distributions supported on alphabets of size $L = 10, 10^2, 10^3, 10^4$ respectively.

While we don't offer a mathematical proof for Conjecture 4.4, tens of thousands of randomly generated numerical experiments suggest that it holds true.

Let us further inspect the proposed estimator and its variance. We first show that it is strictly upper bounded by $\widetilde{\text{KL}}_n$.

**Lemma 4.5.** *For any realization of $\varepsilon_n$, we have*

$$0 \leq \widehat{D_{KL}}(\varepsilon_n) \leq \widetilde{KL}_n, \qquad (26)$$

*and hence*

$$E_{\boldsymbol{y} \sim \pi^{(n)}(\cdot|\boldsymbol{x})}\left[\widehat{D_{KL}}(\varepsilon_n)\right] \leq \widetilde{KL}_n. \qquad (27)$$

Given this, we can immediately bound the variance of the estimator too. Lemma 4.5 implies that standard deviation of the estimator is upper bounded by $\log n$, which in turn implies that if the estimator is averaged over $M = O(\log n \log \frac{1}{\delta})$ draws from the best-of-$n$ model, the standard deviation is guaranteed to be smaller than $\delta$. Given that we are generally interested in $n < 1000$, the dependence on $n$ is mild. Having said that, given each of the $M$ batches contains $n$ iid samples (total of $M \times n$ iid samples), one should be able to build a bootstrapped estimator for the variance with better guarnatees.

In what follows we numerically inspect the proposed estimator in a few scenarios, and compare it with the analytical formula and the exact KL divergence between the best-of-$n$ policy and the reference policy.

The first set of examples, in Figure 2, are uniform distributions over alphabets of varying sizes. Notice that $\varepsilon_n = \varepsilon_\infty = \frac{1}{L}$ for a uniform distribution, and hence the estimator in Equation (25) and Equation (23) are deterministic. As can be seen KL divergence saturates around

*Figure 3.* The analytical formula $(\log(n) - (n-1)/n)$, Equation (4), the alternate bound, Equation (23), the proposed estimator, Equation (25), and the exact KL divergence, for two cherry picked examples. In the left panel, the output is supported on an alphabet of size 5, where the highest reward outcome has a probability of $10^{-4}$ and the rest of the probability mass is uniformly distributed over the rest of the outcomes. In the right panel, the output is supported on an alphabet of size 200, where the three highest reward outcomes have probabilities $10^{-5}, 10^{-3}$ and $10^{-1}$ respectively. The rest of the probability mass is uniformly distributed over the rest of the outcomes.

$n \approx L$. For $\frac{n}{L} \ll 1$, the analytical formula of Equation (4), $\log(n) - (n-1)/n$, has a small gap with the actual KL divergence (which was also theoretically established in Corollary 3.5). On the other hand, when $\frac{n}{L} \gg 1$, the gap between $\log(n) - (n-1)/n$ and the actual KL divergence becomes large and unbounded (which was also theoretically establish in Corollary 3.7). The alternate bound in Equation (23) captures the behavior of the KL divergence for $\frac{n}{L} \gg 1$ and has a finite gap. However, it has a gap of $(n-1)/n$ for $\frac{n}{L} \ll 1$ as previously discussed. Finally, we also observe that the proposed estimator in Equation (25) follows the behavior of the true KL divergence closely in all examples.

In the second set of examples, we cherry pick the probability mass function on the outcome to create different behaviors of the KL divergence, shown in Figure 3. In the left panel, the output is supported on an alphabet of size 5, where the highest reward outcome has a probability of $10^{-4}$ and the rest of the probability mass is uniformly distributed over the rest of the outcomes. We observe that KL divergence saturates early until the highest reward outcome is discovered with $n \approx 10^4$. In the right panel, the output is supported on an alphabet of size 200, where the highest reward outcome has a probability of $10^{-5}$, the second highest reward outcome has a probability of $10^{-3}$, and the third highest reward outcome has a probability of $10^{-1}$. The rest of the probability mass is uniformly distributed over the rest of the outcomes. As can be seen, the KL divergence starts to saturate until the next high reward is outcome is discovered around $n \approx 10^3$ and $n \approx 10^5$. As can be seen, the analytical formula in Equation (4) does not capture the behavior of the KL divergence at all whereas the alternate bound in Equation (23) is much better aligned with the actual behavior. Finally, we observe that the proposed estimator in Equation (25) closely follows the actual KL divergence. We would like to recall that the proposed estimator is random

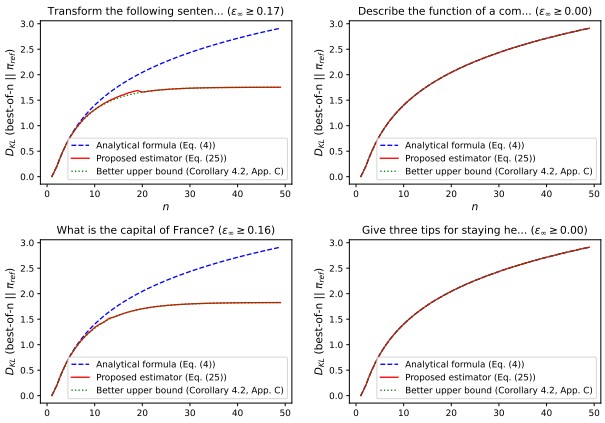

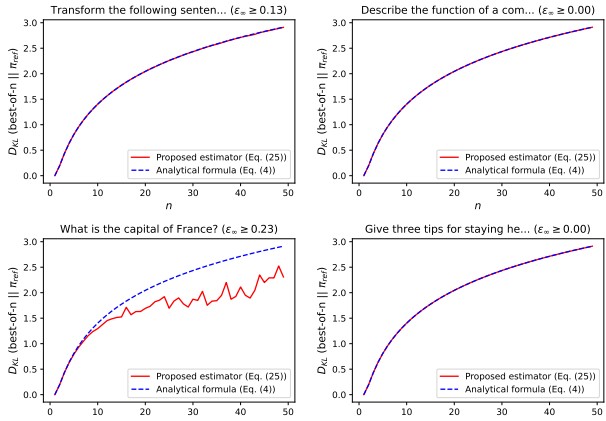

*Figure 4.* The analytical formula $(\log(n) - (n-1)/n)$, Equation (4), better upper bound (Corollary 4.2, Appendix C), the proposed estimator, Equation (25), and the exact KL divergence, for four cherry picked examples from the Alpaca dataset (Taori et al., 2023) using Gemma 9B IT model (Gemma et al., 2024) with reward the log-likelihood of response under the reference model.

*Figure 5.* The analytical formula $(\log(n) - (n-1)/n)$, Equation (4), better upper bound (Corollary 4.2, Appendix C), the proposed estimator, Equation (25), and the exact KL divergence, for four cherry picked examples from the Alpaca dataset (Taori et al., 2023) using Gemma 9B IT model (Gemma et al., 2024) with reward the negative length of the response.

and we have plotted the expected value of the estimators, whereas in practice both estimators are subject to variance due to the randomness in the value of $\varepsilon_n$ (Equation (19)). To capture the deviation from mean, we compute the *tilted min* and *tilted max* (see Appendix C.2 for the definition), which are also plotted in Figure 3. As can be seen, in cases where $\varepsilon_n$ could widely vary based on whether a certain outcome appears in the set of $n$ outcomes, the variance could be high.

In Figure 4, we compare the estimates for four cherry picked examples from the Alpaca dataset (Taori et al., 2023) using Gemma 9B IT model (Gemma et al., 2024) with reward being the log-likelihood of the reference model. Note that for two examples where $\varepsilon_\infty$ is large, i.e., the prompts that induce less entropy in the response, such as *"What is the capital of France?"*, the proposed estimator outperforms the analytical formula in Equation (4) considerably and lies very close to the better upper bound in Corollary 4.2, whereas $\widetilde{\mathrm{KL}}_n$ (Equation (4)) is loose even for $n \approx 20$. The details on the prompts used can be found in Appendix C. In Figure 5, we repeat the same experiment but change the reward to the negative of length to prefer more concise responses and see similar trends. We also include experiments with machine translation in Appendix C.3.

## 5. Win Rate of the Best-of-n Policy

So far, we provided theoretical guarantees on the KL divergence of the best-of-$n$ policy with respect to the reference policy. In this section, we extend our study to characterize the *win rate* of the best-of-$n$ policy against the reference policy. Let *win* of $\boldsymbol{y}$ against $\boldsymbol{z}$ in context $\boldsymbol{x}$ be defined as:

$$w_r(\boldsymbol{y} \succ \boldsymbol{z}|\boldsymbol{x}) := \mathbf{1}(r(\boldsymbol{x}, \boldsymbol{y}) > r(\boldsymbol{x}, \boldsymbol{z})) + \frac{\mathbf{1}(r(\boldsymbol{x}, \boldsymbol{y}) = r(\boldsymbol{x}, \boldsymbol{z}))}{2}.$$

In other words, $w_r(\boldsymbol{y} \succ \boldsymbol{z}|\boldsymbol{x})$ indicates whether response $\boldsymbol{y}$ wins over response $\boldsymbol{z}$ in context $\boldsymbol{x}$ using the judge $r$. Then, let the *win rate* of policy $\pi$ over the reference policy $\pi_{\mathrm{ref}}$ for prompt $\boldsymbol{x}$ be defined as

$$\mathcal{W}_r(\pi(\cdot|\boldsymbol{x})\|\pi_{\mathrm{ref}}(\cdot|\boldsymbol{x})) := E_{\boldsymbol{y}\sim\pi(\cdot|\boldsymbol{x})} E_{\boldsymbol{z}\sim\pi_{\mathrm{ref}}(\cdot|\boldsymbol{x})} w_r(\boldsymbol{y} \succ \boldsymbol{z}|\boldsymbol{x}). \tag{28}$$

It is clear that $\mathcal{W}_r(\pi_{\mathrm{ref}}(\cdot|\boldsymbol{x})\|\pi_{\mathrm{ref}}(\cdot|\boldsymbol{x})) = 0.5$ and the goal of alignment is to improve win rate beyond $0.5$ (with the lowest KL divergence between the two models). We further define the following, averaged over prompt distribution $\mu$ :

$$\mathcal{W}_r^\mu(\pi\|\pi_{\mathrm{ref}}) := E_{\boldsymbol{x}\sim\mu}\mathcal{W}_r(\pi(\cdot|\boldsymbol{x})\|\pi_{\mathrm{ref}}(\cdot|\boldsymbol{x})). \tag{29}$$

Note that it is clear that if $\boldsymbol{y} \sim \pi(\cdot|\boldsymbol{x})$ and $\boldsymbol{z} \sim \pi_{\mathrm{ref}}(\cdot|\boldsymbol{x})$, then $w_r(\boldsymbol{y} \succ \boldsymbol{z}|\boldsymbol{x})$ is an unbiased estimator for win rate of policy $\pi$ against $\pi_{\mathrm{ref}}$.

We analyze the win rate and derive theoretical guarantees. Let $\mathcal{F}_\pi$ and $\mathcal{F}_\pi^-$ be defined in Equation (7) and Equation (8), respectively. We define *calibrated reward* as:

$$\mathcal{C}_{\pi_{\mathrm{ref}}}(\boldsymbol{x}, \boldsymbol{y}) := \frac{\mathcal{F}_{\pi_{\mathrm{ref}}}(\boldsymbol{y}|\boldsymbol{x}) + \mathcal{F}_{\pi_{\mathrm{ref}}}^-(\boldsymbol{y}|\boldsymbol{x})}{2}. \tag{30}$$

With this definition in place, notice that win rate could be expressed as follows.

**Lemma 5.1.** *The win rate of any policy $\pi$ against reference policy $\pi_{ref}$ could be expressed as*

$$\mathcal{W}_r(\pi(\cdot|\boldsymbol{x})\|\pi_{ref}(\cdot|\boldsymbol{x})) = E_{\boldsymbol{y}\sim\pi(\cdot|\boldsymbol{x})} \left[ \mathcal{C}_{\pi_{ref}}(\boldsymbol{x}, \boldsymbol{y}) \right]. \tag{31}$$

Notice that the above result suggests that the win rate of any policy only depends on reward and the reference policy through the calibrated reward function, $\mathcal{C}_{\pi_{\mathrm{ref}}}(\boldsymbol{x}, \boldsymbol{y})$. Hence, this notion of calibration may be used as a canonical transformation of the reward for preference optimization against

a given reference policy. In fact, this transformation is theoretically proposed as the objective in IPO (Azar et al., 2024) and is the key to best-of-$n$ distillation (Gui et al., 2024; Amini et al., 2025; Sessa et al., 2025; Yang et al., 2024b) and inference-aware alignment (Balashankar et al., 2025).

**Lemma 5.2.** *The win rate of best-of-n policy against $\pi_{ref}$ is given by*

$$\mathcal{W}_r(\pi^{(n)}(\cdot|\boldsymbol{x})\|\pi_{ref}(\cdot|\boldsymbol{x}))$$
$$= \sum_{\boldsymbol{y}} \left( \mathcal{F}_{\pi_{ref}}(\boldsymbol{y}|\boldsymbol{x})^n - \mathcal{F}_{\pi_{ref}}^{-}(\boldsymbol{y}|\boldsymbol{x})^n \right) \mathcal{C}_{\pi_{ref}}(\boldsymbol{x},\boldsymbol{y}).$$

*Proof.* This is proved by plugging Lemma 2.3 into Lemma 5.1. $\square$

Our next result is an upper bound on the win rate of best-of-$n$ policy. Intuitively, observe that the win rate could be estimated by drawing $(n+1)$ samples from $\pi_{ref}$, associating $n$ to the best-of-$n$ model and the remaining one to the reference model. Hence, the best-of-$n$ model gets $n$-to-1 chances of winning against the reference model, unless there is a draw due to samples with the same reward value. Thus, intuitively $\mathcal{W}_r(\pi^{(n)}(\cdot|\boldsymbol{x})\|\pi_{ref}(\cdot|\boldsymbol{x})) \approx \frac{n}{n+1}$. We formalize this as an upper bound on the win rate.

**Theorem 5.3.** *For all, $n$, and all $\boldsymbol{x}$, the win rate of best-of-n policy is upper bounded by*

$$\mathcal{W}_r(\pi^{(n)}(\cdot|\boldsymbol{x})\|\pi_{ref}(\cdot|\boldsymbol{x})) \leq \frac{n}{n+1}. \quad (32)$$

In the rest of this section, we derive bounds on the gap between this upper bound and the actual win rate:

$$G_{\mathcal{W}}^{(n)}(\boldsymbol{x}) := \frac{n}{n+1} - \mathcal{W}_r(\pi^{(n)}(\cdot|\boldsymbol{x})\|\pi_{ref}(\cdot|\boldsymbol{x})) \geq 0. \quad (33)$$

### 5.1. Upper Bounds on the Win Rate Gap

Unlike the KL divergence, it is clear that $G_{\mathcal{W}}^{(n)}(\boldsymbol{x})$ could not grow unbounded, and is upper bounded by $\frac{1}{2}$.

**Theorem 5.4.** *The win rate gap is upper bounded by*

$$G_{\mathcal{W}}^{(n)}(\boldsymbol{x}) \leq \frac{n-1}{2} e^{-H_2(\pi_{ref}|\boldsymbol{x})}, \quad (34)$$

*where $H_2(\cdot)$ denotes the Rényi entropy of order 2 defined in Equation (16).*

**Corollary 5.5.** *Let $\pi_{ref}(\boldsymbol{y}|\boldsymbol{x}) \leq \delta$ for all $\boldsymbol{y} \in \mathcal{Y}^*$, i.e., $\pi_{ref}$ is $\delta$-bound. Then,*

$$G_{\mathcal{W}}^{(n)}(\boldsymbol{x}) \leq \frac{n-1}{2}\delta. \quad (35)$$

*Proof.* The proof follows by noticing that $H_2(\pi_{ref}|\boldsymbol{x}) \geq \log(1/\delta)$ and invoking Theorem 5.4. $\square$

Given this upper bound, the win rate of best-of-$n$ would be fairly close to $\frac{n}{n+1}$ if $\pi_{ref}$ is a $\delta$-bound model, and $n$ is sufficiently small such that $n\delta \ll 1$, by combining Theorem 5.3 and Corollary 5.5, we get

$$\mathcal{W}_r(\pi^{(n)}(\cdot|\boldsymbol{x})\|\pi_{ref}(\cdot|\boldsymbol{x})) \approx \frac{n}{n+1}, \quad (36)$$

which is the claim of Gui et al. (2024, Theorem 2) for the win rate of best-of-$n$

### 5.2. Lower Bounds on the Win Rate Gap

Our next result characterizes the cases where the gap is bounded away from 0.

**Theorem 5.6.** *Let $\varepsilon_\infty > 0$ be defined in Equation (20). Then,*

$$G_{\mathcal{W}}^{(n)}(\boldsymbol{x}) \geq \frac{n}{n+1}(1-(1-\varepsilon_\infty)^{n+1})$$
$$- (1-(1-\varepsilon_\infty)^n)\left(1-\frac{\varepsilon_\infty}{2}\right) > 0.$$

**Corollary 5.7.** *As $n \to \infty$, we have*

$$G_{\mathcal{W}}^{(n)}(\boldsymbol{x}) \geq \frac{\varepsilon_\infty}{2}(1+o_n(1)). \quad (37)$$

As $n \to \infty$, $G_{\mathcal{W}}^{(n)}(\boldsymbol{x}) \to \frac{\varepsilon_\infty}{2}$, which is bounded from 0.

## 6. Rewind-and-Repeat: Rejection Sampling Beyond Best-of-n

The best-of-$n$ policy is a form of rejection sampling. Another form is called rewind-and-repeat, where the process of generating a response and scoring it is repeated until a certain threshold on reward is met (Kim et al., 2025). A more involved blockwise variant of this process is recently used by Li et al. (2024). Formally, let $\boldsymbol{x}$ be a given input prompt to the model, and let $\{\boldsymbol{y}_k\}_{k=1}^\infty$ be a sequence of infinite i.i.d. samples drawn from $\pi_{ref}(\cdot|\boldsymbol{x})$. Then, rewind-and-repeat accepts $\boldsymbol{y}_M$ such that

$$r(\boldsymbol{x},\boldsymbol{y}_M) \geq \Phi \quad \text{and} \quad \forall k < M : r(\boldsymbol{x},\boldsymbol{y}_k) < \Phi, \quad (38)$$

where $\Phi \in \mathbb{R}$ is the threshold on reward. In other words, $\boldsymbol{y}_M$ is the first draw whose reward reaches a certain threshold $\Phi$. We also call $M$ the (random) number of trials until the threshold is met, which determines the cost of inference from the model. We denote the resulting policy by $\pi_\Phi$.

It is natural to ask: *how do the win rate vs KL tradeoffs of rewind-and-repeat compare with that of best-of-n?* To answer this question, first we define

$$w_\Phi(\boldsymbol{x}) := E_{\boldsymbol{y}\sim\pi_{ref}(\cdot|\boldsymbol{x})}[\mathbf{1}(r(\boldsymbol{x},\boldsymbol{y}) \geq \Phi)] \quad (39)$$

as the probability of drawing a sample from the reference policy that meets the threshold. Hence, the expected number of trials to output an outcome is $E[M] = 1/w_\Phi(\boldsymbol{x})$.

Next, let us derive the PMF of rewind-and-repeat policy.

**Lemma 6.1.** *The probability mass function (PMF) of the rewind-and-repeat policy is given by*

$$\pi_{\Phi}(\boldsymbol{y}|\boldsymbol{x}) = \begin{cases} \frac{\pi_{ref}(\boldsymbol{y}|\boldsymbol{x})}{w_{\Phi}(\boldsymbol{x})} & \text{if } r(\boldsymbol{x}, \boldsymbol{y}) \geq \Phi \\ 0 & \text{if } r(\boldsymbol{x}, \boldsymbol{y}) < \Phi \end{cases} . \quad (40)$$

The proofs are deferred to Appendix A.4. Given its PMF, next we derive KL divergence and win rate of the rewind-and-repeat policy and the reference policy.

**Lemma 6.2.** *We have*

$$D_{KL}(\pi_{\Phi}(\cdot|\boldsymbol{x})\|\pi_{ref}(\cdot|\boldsymbol{x})) = \log \frac{1}{w_{\Phi}(\boldsymbol{x})}, \quad (41)$$

$$\mathcal{W}_r(\pi_{\Phi}(\cdot|\boldsymbol{x})\|\pi_{ref}(\cdot|\boldsymbol{x})) = 1 - \frac{1}{2}w_{\Phi}(\boldsymbol{x}). \quad (42)$$

Thus, by sweeping $\Phi$ in $\mathbb{R}$, we will effectively sweep $w_{\Phi}$ in $[0, 1]$, and obtain the respective win rate vs KL divergence tradeoff for the rewind-and-repeat policy. Note that the KL divergence was recently derived by Kim et al. (2025, Appendix A.5) and we provide a proof for completeness.

So far, we derived a characterization of the KL divergence of the rewind-and-repeat policy. However, when the number of outcomes of a model is large, similarly to the case of best-of-$n$, estimating the KL divergence is intractable.

Our main result in this section is an unbiased estimator of the KL divergence of the rewind-and-repeat procedure.

**Theorem 6.3.** *For $n \geq 1$, let $H_n := \sum_{i=1}^{n} \frac{1}{i}$ be the $n$-th Harmonic number and $H_0 = 0$. Further, let $M$ be the number of trials to achieve an outcome in the rewind-and-repeat policy. Then,*

$$D_{KL}(\pi_{\Phi}(\cdot|\boldsymbol{x})\|\pi_{ref}(\cdot|\boldsymbol{x})) = E[H_{M-1}]. \quad (43)$$

Hence, we propose $H_{M-1}$ as an (unbiased) estimator for the KL divergence of the rewind-and-repeat procedure with respect to the reference policy.

# 7. Win rate vs KL Divergence Tradeoffs

Thus far, we characterized the KL divergence and win rate of best-of-$n$. In practice, it is customary to compare different alignment methods based on their win rate at a certain KL divergence from the reference policy. Note that Theorem 3.1 implies that the win rate (or expected reward) vs KL tradeoffs reported in the literature that use the analytical formula in Equation (4) (Gao et al., 2023; Go et al., 2023; Mudgal et al., 2024; Scheurer et al., 2023) are conservative and the actual tradeoff curve of the best-of-$n$ policy is in fact guaranteed to be no worse than what is reported. To further substantiate this point, let us revisit Example 1 and report the win rate vs KL divergence tradeoff curve (Figure 6), where we used the actual win rate in all cases.[4] The actual

---

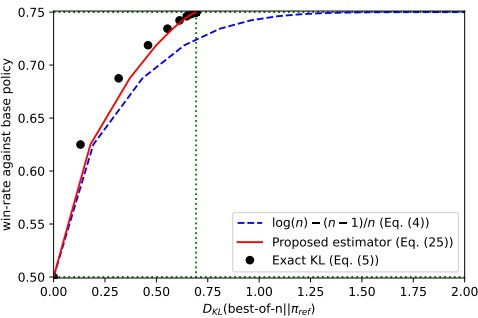

*Figure 6.* The win rate against reference policy vs KL divergence tradeoff curve for the best-of-$n$ policy. We have used the analytical formula $(\log(n) - (n - 1)/n)$, Equation (4), the exact KL divergence, Equation (5), and the proposed estimator, Equation (25), for producing the tradeoffs from Example 1, illustrating a case where the actual win rate vs KL divergence tradeoff curve for the best-of-$n$ policy is more favorable than the one predicted if using the upper bound formula, $\log(n) - (n - 1)/n$.

win rate vs KL divergence tradeoff is more favorable than that portrayed by using the formula in Equation (4). In this example, the best-of-$n$ policy in the limit of large $n$, reaches a KL divergence of $\log(2)$ and a win rate of $0.75$.

**Definition 7.1.** Let $W : \mathbb{R}^+ \to [0, 1]$ be a function that takes in $D \geq 0$ and outputs $W(D)$. Let $D_n = D_{\mathrm{KL}}(\pi^{(n)}(\cdot|\boldsymbol{x})\|\pi_{\mathrm{ref}}(\cdot|\boldsymbol{x}))$. We say that $W$ *is an upper bound on the tradeoff curve of best-of-$n$* if for all $n$, $\mathcal{W}_r(\pi^{(n)}(\cdot|\boldsymbol{x})\|\pi_{\mathrm{ref}}(\cdot|\boldsymbol{x})) \leq W(D_n)$. Alternatively, we say that $W$ *is a lower bound on the tradeoff curve of best-of-$n$* if for all $n$, $\mathcal{W}_r(\pi^{(n)}(\cdot|\boldsymbol{x})\|\pi_{\mathrm{ref}}(\cdot|\boldsymbol{x})) \geq W(D_n)$.

We can turn our existing bounds into straightforward lower and upper bounds on the tradeoff curve for best-of-$n$. (see Appendix A.5). We conjecture a tighter upper bound.

**Conjecture 7.2.** *For any $\boldsymbol{x}$ and a given $\pi_{ref}$, let the function $W_0(D) = \ell^{-1}(D)$ where for all $\tau \in [0.5, 1)$,*

$$\ell(\tau) = \log \frac{\tau}{1 - \tau} + \frac{1}{\tau} - 2.$$

*$W_0$ is an upper bound on the tradeoff curve of best-of-$n$.*

**Example 2.** We consider a ternary language model with alphabet $\mathcal{X} = \{0, 1, 2\}$, ordered from least preferred to most preferred, with probabilities given by $\pi_{\mathrm{ref}} = (0.3, 0.6, 0.1)$. Hence, the calibrated reward in Equation (30) is given by $(0.3, 0.75, 0.95)$. The set of solutions to the KL-regularized RL problem are given by Equation (2). We also compute the best-of-$n$ solutions (for continuous $n$) using Lemma 2.3 and rewind-and-repeat using Lemma 6.1. Before discussing the win rate vs KL divergence tradeoffs, we first visualize the set of solutions on the probability simplex in Figure 7 and observe that the solutions could be very different. When we consider the win rate vs KL tradeoffs for this example (Figure 8), we observe that the tradeoffs are strikingly close

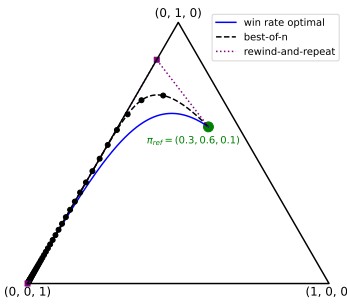

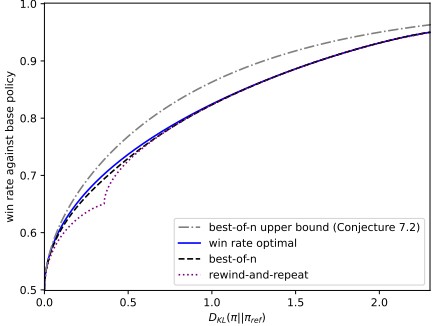

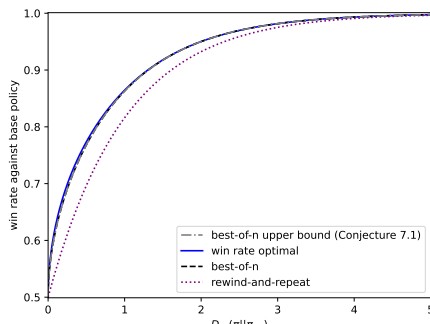

*Figure 7.* The set of solutions for Example 2: *win rate optimal* is achieved by varying $\beta$ in Equation (2), *best-of-n* is given by Lemma 2.3, and *rewind-and-repeat* is given by Lemma 6.1.

*Figure 8.* Win rate vs KL tradeoff for Example 2. The tradeoff curve of *Best-of-n* is close to *win rate optimal*, and both are better than rewind-and-repeat.

*Figure 9.* Win rate vs KL tradeoff for Example 3. The tradeoff curve of *Best-of-n* is close to *win rate optimal* and the limit behavior, and both are better than rewind-and-repeat.

for best-of-$n$ and win rate optimal models, matching recent findings of Yang et al. (2024a); Gui et al. (2024).

**Example 3.** We consider a language model with low probability outcomes, and a calibrated reward, and plot win rate vs KL divergence tradeoffs in Figure 9, and observe that in this asymptotic regime where we observe that the tradeoff curve offered by best-of-$n$ is almost optimal and offers better tradeoffs compared to rewind-and-repeat.

## 8. Conclusion

We studied the best-of-$n$ alignment policy and derived its probability mass function (Lemma 2.3). We proved that an analytical formula used in the literature for the KL divergence of the best-of-$n$ policy with the reference policy, $\log(n) - (n-1)/n$ in Equation (4), is false, and only an upper bound on the KL divergence (Theorem 3.1). We derived bounds on the gap between this formula and the KL divergence where we roughly showed the following: Let $\boldsymbol{y}$ be a draw from the best-of-$n$ policy. Let $\varepsilon_n$ be the probability mass of $\boldsymbol{y}$ under the base model. Then, if $n\varepsilon_n \ll 1$, the gap between the formula in Equation (4) and the exact KL divergence is small (Theorem 3.4); and if $n\varepsilon_n \gg 1$, the gap between the two may be large and unbounded (Theorem 3.6). We proposed a new estimator for the KL divergence (Definition 4.3), which we demonstrated to capture the behavior of the KL divergence on several numerical experiments. We showed that the win rate of best-of-$n$ against the reference policy is upper bounded by $n/(n+1)$ (Theorem 5.3). Similarly to the KL divergence, we provided upper (Theorem 5.4) and lower (Theorem 5.6) bounds on the gap between the actual win rate and the bound. We compared best-of-$n$ with another form of rejection sampling through rewind-and-repeat and showed its superiority both theoretically and empirically. We also extended the bounds to blockwise best-of-$n$ (Mudgal et al., 2024).

While our results showed that best-of-$n$ offers better trade-

offs on reward vs KL divergence (where KL divergence captures preservation of the core model capabilities) compared to rewind-and-repeat, the latter is more effective in driving reward up for a given compute budget which is important in test-time compute scaling. Hence, it remains to be seen how to best design a method that achieves Pareto optimal tradeoffs between compute, reward, and KL divergence (which captures preservation of capabilities other than what is captured by reward). This might involve combining the rewind-and-repeat with best-of-$n$ and could be an area for future work.

## Impact Statement

This paper presents theoretical investigations of best-of-$n$ sampling and other test-time rejection sampling algorithms, which is a simple yet effective method for test-time alignment of generative models. One of the major findings of this paper is that a widely used formula for KL divergence of the best-of-$n$ policy and the reference policy is a theoretical upper bound on the actual KL divergence. This may help ensure that the capabilities of the reference model are largely preserved in the aligned model. For example, this may help compliance or risk-management teams preserve safety by guaranteeing that the policy that is served does not drift too far from a safe reference policy.

Our work also showed that best-of-$n$ is an effective (and almost optimal) test-time alignment method which comes with theoretical guarantees on win rate vs KL divergence tradeoffs motivating its use for improving the capabilities and safety of models. On the flip side, this also shows why best-of-$n$ with an adversarial reward may be used to effectively jailbreak generative models. We hope future work can benefit from our findings in making best-of-$n$ more effective, and can also devise safeguards against best-of-$n$ jailbreaks.

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

# A. Proofs of the main results of the paper

## A.1. Proofs of Section 3

We provide the proofs of the main results of the paper. To do so, we need to state two further auxiliary lemmas.

**Lemma A.1.** *For any $0 \leq a < b \leq 1$, and $n \in \mathbb{N}$,*

$$(b^n - a^n) \log \frac{b^n - a^n}{b - a} = \int_a^b nv^{n-1} \log\left(nv^{n-1}\right) dv - g_n(a, b) \leq \int_a^b nv^{n-1} \log\left(nv^{n-1}\right) dv, \tag{44}$$

*where $g(a, b) \geq 0$, and is given by*

$$g_n(a, b) := (b^n - a^n) D_{KL}(p_v \| p_u) = (b^n - a^n) \log \frac{n(b - a)}{b^n - a^n} + (n - 1)(b^n \log b - a^n \log a) - \frac{n - 1}{n}(b^n - a^n), \tag{45}$$

*where for $a \leq w \leq b$, we define $p_v(w) = \frac{nw^{n-1}}{b^n - a^n}$ and $p_u(w) = \frac{1}{b - a}$.*

*Proof.* Notice that

$$D_{KL}(p_v \| p_u) = \int_a^b \frac{nv^{n-1}}{b^n - a^n} \log \frac{\frac{nv^{n-1}}{b^n - a^n}}{\frac{1}{b - a}} dv = \int_a^b \frac{nv^{n-1}}{b^n - a^n} \log \frac{nv^{n-1}(b - a)}{b^n - a^n} dv \geq 0. \tag{46}$$

The inequality is established by algebraic manipulation. The gap $g_n(a, b)$ is obtained by following Lemma A.2 to derive the right-hand-side of the inequality in closed form. $\square$

**Lemma A.2.** *The following identity holds:*

$$\int_a^b nv^{n-1} \log\left(nv^{n-1}\right) dv = (b^n - a^n) \log n + (n - 1)\left(b^n \log b - a^n \log a\right) - \frac{n - 1}{n}(b^n - a^n). \tag{47}$$

*Proof.* The proof is completed by noticing that $\frac{d}{dv}(v^n \log v) = nv^{n-1} \log v + v^{n-1}$. $\square$

*Proof of Theorem 3.1.*

$$D_{KL}(\pi^{(n)}(\cdot|\boldsymbol{x}) \| \pi_{\text{ref}}(\cdot|\boldsymbol{x})) = \sum_{\boldsymbol{y} \in \mathcal{Y}^*} \left(\mathcal{F}_{\pi_{\text{ref}}}(\boldsymbol{y}|\boldsymbol{x})^n - \mathcal{F}_{\pi_{\text{ref}}}^-(\boldsymbol{y}|\boldsymbol{x})^n\right) \log \frac{\mathcal{F}_{\pi_{\text{ref}}}(\boldsymbol{y}|\boldsymbol{x})^n - \mathcal{F}_{\pi_{\text{ref}}}^-(\boldsymbol{y}|\boldsymbol{x})^n}{\pi_{\text{ref}}(\boldsymbol{y}|\boldsymbol{x})} \tag{48}$$

$$\leq \sum_{\boldsymbol{y} \in \mathcal{Y}^*} \int_{\mathcal{F}_{\pi_{\text{ref}}}^-(\boldsymbol{y}|\boldsymbol{x})}^{\mathcal{F}_{\pi_{\text{ref}}}(\boldsymbol{y}|\boldsymbol{x})} nv^{n-1} \log\left(nv^{n-1}\right) dv \tag{49}$$

$$= \int_0^1 nv^{n-1} \log\left(nv^{n-1}\right) dv \tag{50}$$

$$= \log(n) - \frac{n - 1}{n}, \tag{51}$$

where Equation (49) follows from Lemma A.1, and Equation (51) follows from Lemma A.2. $\square$

**Lemma A.3.** *For any $0 \leq a < b \leq 1$, and $n \in \mathbb{N}$,*

$$g_n(a, b) \leq 2n(n - 1)(b - a)^2. \tag{52}$$

*Proof.* Recall from Equation (45) that

$$g_n(a, b) := (b^n - a^n) D_{\text{KL}}(p_v \| p_u), \tag{53}$$

where $p_v(w) = \frac{nw^{n-1}}{b^n - a^n}$ and $p_u(w) = \frac{1}{b-a}$ for $a \le w \le b$. We can further bound the KL divergence as

$$D_{\text{KL}}(p_v \| p_u) \le \max_{w \in [a,b]} \log \frac{p_v(w)}{p_u(w)} \tag{54}$$

$$= \max_{w \in [a,b]} \log \frac{nw^{n-1}(b-a)}{b^n - a^n} \tag{55}$$

$$= \log \frac{nb^{n-1}(b-a)}{b^n - a^n} \tag{56}$$

$$= \log \frac{nb^{n-1}}{\sum_{j=0}^{n-1} b^{n-1-j} a^j}. \tag{57}$$

At this point, we will divide the proof into two cases depending on the value of $a, b$.

*Case I.* We first focus on the case when $a < b/2$. In this case,

$$D_{\text{KL}}(p_v \| p_u) \le \log \frac{nb^{n-1}}{\sum_{j=0}^{n-1} b^{n-1-j} a^j} \tag{58}$$

$$\le \log n. \tag{59}$$

Hence,

$$g_n(a, b) \le (b^n - a^n) \log n \tag{60}$$

$$\le b^n \log n \tag{61}$$

$$\le b^2 \log n \tag{62}$$

$$\le 4(b-a)^2 \log n \tag{63}$$

$$\le 2n(n-1)(b-a)^2. \tag{64}$$

*Case II.* For $a \ge b/2$, notice that

$$D_{\text{KL}}(p_v \| p_u) \le \log \frac{nb^{n-1}}{\sum_{j=0}^{n-1} b^{n-1-j} a^j} \tag{65}$$

$$\le \log \frac{b^{n-1}}{b^{(n-1)/2} a^{(n-1)/2}} \tag{66}$$

$$= \frac{n-1}{2} \log \frac{b}{a} \tag{67}$$

$$\le \frac{n-1}{2} \frac{(b-a)}{a} \tag{68}$$

$$\le (n-1) \frac{(b-a)}{b}, \tag{69}$$

where the first inequality follows from AM-GM inequality. Hence,

$$g_n(a, b) \le (b^n - a^n)(n-1) \frac{(b-a)}{b} \tag{70}$$

$$= (b-a) \left( \sum_{j=0}^{n-1} b^{n-1-j} a^j \right) (n-1) \frac{(b-a)}{b} \tag{71}$$

$$\le nb^{n-1}(b-a)(n-1) \frac{(b-a)}{b} \tag{72}$$

$$= nb^{n-2}(b-a)(n-1)(b-a) \tag{73}$$

$$\le 2n(n-1)(b-a)^2. \tag{74}$$

The proof is completed by putting together *Case I* and *Case II*. □

*Proof of Theorem 3.4.*

$$\log(n) - \frac{n-1}{n} - D_{\text{KL}}(\pi^{(n)}(\cdot|\boldsymbol{x})\|\pi_{\text{ref}}(\cdot|\boldsymbol{x})) = \sum_{\boldsymbol{y}\in\mathcal{Y}^*} g_n(\mathcal{F}^-_{\pi_{\text{ref}}}(\boldsymbol{y}|\boldsymbol{x}), \mathcal{F}_{\pi_{\text{ref}}}(\boldsymbol{y}|\boldsymbol{x})) \tag{75}$$

$$\leq 2n(n-1)\sum_{\boldsymbol{y}\in\mathcal{Y}^*} (\mathcal{F}_{\pi_{\text{ref}}}(\boldsymbol{y}|\boldsymbol{x}) - \mathcal{F}^-_{\pi_{\text{ref}}}(\boldsymbol{y}|\boldsymbol{x}))^2 \tag{76}$$

$$= 2n(n-1)\sum_{\boldsymbol{y}\in\mathcal{Y}^*} (\pi_{\text{ref}}(\boldsymbol{y}|\boldsymbol{x}))^2 \tag{77}$$

$$= 2n(n-1)e^{-H_2(\pi_{\text{ref}}|\boldsymbol{x})}. \tag{78}$$

where Equation (76) follows from Lemma A.3. □

*Proof of Theorem 3.6.* Notice that the gap is at least lower bounded by the value of the gap for the highest reward outcome. Hence,

$$\log(n) - \frac{n-1}{n} - D_{\text{KL}}(\pi^{(n)}(\cdot|\boldsymbol{x})\|\pi_{\text{ref}}(\cdot|\boldsymbol{x})) \geq g_n(1-\varepsilon_\infty, 1), \tag{79}$$

where $g_n(\cdot, \cdot)$ is defined in Equation (45), and is given by

$$g_n(1-\varepsilon_\infty, 1) = (1 - (1-\varepsilon_\infty)^n)\log\frac{n\varepsilon_\infty}{1 - (1-\varepsilon_\infty)^n} - (n-1)(1-\varepsilon_\infty)^n\log(1-\varepsilon_\infty) - \frac{n-1}{n}(1 - (1-\varepsilon_\infty)^n), \tag{80}$$

completing the proof. □

*Proof of Corollary 3.7.* The proof is completed by the following inequalities:

$$g_n(1-\varepsilon_\infty, 1) = (1 - (1-\varepsilon_\infty)^n)\log\frac{n\varepsilon_\infty}{1 - (1-\varepsilon_\infty)^n} - (n-1)(1-\varepsilon_\infty)^n\log(1-\varepsilon_\infty) - \frac{n-1}{n}(1 - (1-\varepsilon_\infty)^n) \tag{81}$$

$$\geq (1 - e^{-n\varepsilon_\infty})\left(\log\frac{n\varepsilon_\infty}{1 - (1-\varepsilon_\infty)^n} + (n-1)\frac{(1-\varepsilon_\infty)^n}{1 - (1-\varepsilon_\infty)^n}\log\frac{1}{1-\varepsilon_\infty} - \frac{n-1}{n}\right) \tag{82}$$

$$\geq (1 - e^{-n\varepsilon_\infty})\left(\log\frac{n\varepsilon_\infty}{1 - (1-\varepsilon_\infty)^n} + (n-1)\frac{\varepsilon_\infty(1-\varepsilon_\infty)^n}{1 - (1-\varepsilon_\infty)^n} - \frac{n-1}{n}\right) \tag{83}$$

$$= (1 - e^{-n\varepsilon_\infty})\left(\log\frac{n\varepsilon_\infty}{1 - (1-\varepsilon_\infty)^n} + \frac{n-1}{n}\left(\frac{n\varepsilon_\infty}{1 - (1-\varepsilon_\infty)^n}(1-\varepsilon_\infty)^n - 1\right)\right) \tag{84}$$

$$\geq (1 - e^{-n\varepsilon_\infty})\left(\log(n\varepsilon_\infty) - \frac{n-1}{n}\right) \tag{85}$$

$$\geq (1 - e^{-n\varepsilon_\infty})\log(n\varepsilon_\infty) - 1. \tag{86}$$

Hence, as $n \to \infty$,

$$g_n(1-\varepsilon_\infty, 1) \geq \log(n\varepsilon_\infty) + o_n(\log n), \tag{87}$$

which completes the proof. □

## A.2. Proofs of Section 4

*Proof of Lemma 4.1.* The proof follows from:

$$D_{\text{KL}}(\pi^{(n)}(\cdot|\boldsymbol{x})\|\pi_{\text{ref}}(\cdot|\boldsymbol{x})) = E_{\boldsymbol{y}\sim\pi^{(n)}}\left[\log\left(\frac{\mathcal{F}_{\pi_{\text{ref}}}(\boldsymbol{y}|\boldsymbol{x})^n - \mathcal{F}_{\pi_{\text{ref}}}^{-}(\boldsymbol{y}|\boldsymbol{x})^n}{\pi_{\text{ref}}(\boldsymbol{y}|\boldsymbol{x})}\right)\right] \tag{88}$$

$$\leq E_{\boldsymbol{y}\sim\pi^{(n)}}\left[\log\left(\frac{1 - (1 - \pi_{\text{ref}}(\boldsymbol{y}|\boldsymbol{x}))^n}{\pi_{\text{ref}}(\boldsymbol{y}|\boldsymbol{x})}\right)\right]. \tag{89}$$

$\square$

*Proof of Corollary 4.2.* Recall that $\varepsilon_\infty = \pi_{\text{ref}}(\boldsymbol{y}_{\max}|\boldsymbol{x})$ where $\boldsymbol{y}_{\max} \sim \pi_{\boldsymbol{y}|\boldsymbol{x}}^{(\infty)}$. Notice that

$$D_{\text{KL}}(\pi^{(n)}(\cdot|\boldsymbol{x})\|\pi_{\text{ref}}(\cdot|\boldsymbol{x})) \leq \int_0^{1-\varepsilon_\infty} nv^{n-1}\log\left(nv^{n-1}\right)dv + (1 - (1 - \varepsilon_\infty)^n)\log\frac{1 - (1 - \varepsilon_\infty)^n}{\varepsilon_\infty}, \tag{90}$$

which follows the same lines as in the proof of Theorem 3.1 except that we singled out the highest reward outcome, and bounded the rest of the terms. The proof is completed by invoking Lemma A.2 to express the integral in closed form. $\square$

*Proof of Lemma 4.5.* Fist notice that for any $0 \leq \varepsilon \leq 1$,

$$(1 - (1 - \varepsilon)^n)\log\frac{1 - (1 - \varepsilon)^n}{\varepsilon} \leq \int_{1-\varepsilon}^1 nv^{n-1}\log\left(nv^{n-1}\right)dv, \tag{91}$$

which is implied by Lemma A.1 and setting $b = 1$ and $a = 1 - \varepsilon$. The proof is completed by noticing that

$$\widehat{D_{\text{KL}}}(\varepsilon_n) = \int_0^{1-\varepsilon_n} nv^{n-1}\log\left(nv^{n-1}\right)dv + (1 - (1 - \varepsilon_n)^n)\log\frac{1 - (1 - \varepsilon_n)^n}{\varepsilon_n}, \tag{92}$$

and

$$\widetilde{\text{KL}}_n = \int_0^{1-\varepsilon_n} nv^{n-1}\log\left(nv^{n-1}\right)dv + \int_{1-\varepsilon_n}^1 nv^{n-1}\log\left(nv^{n-1}\right)dv. \tag{93}$$

$\square$

## A.3. Proofs of Section 5

*Proof of Lemma 5.1.* Recall the definition of $\mathcal{F}_\pi$ and $\mathcal{F}_\pi^-$, hence

$$E_{\boldsymbol{z}\sim\pi_{\mathrm{ref}}(\cdot|\boldsymbol{x})}\left[\mathcal{W}_r(\boldsymbol{y}\succ\boldsymbol{z}|\boldsymbol{x})\right] = E_{\boldsymbol{z}\sim\pi_{\mathrm{ref}}(\cdot|\boldsymbol{x})}\left\{\mathbf{1}(r(\boldsymbol{x},\boldsymbol{y})>r(\boldsymbol{x},\boldsymbol{z}))+\frac{1}{2}\mathbf{1}(r(\boldsymbol{x},\boldsymbol{y})=r(\boldsymbol{x},\boldsymbol{z}))\right\}$$

$$= \frac{1}{2}\left(\mathcal{F}_{\pi_{\mathrm{ref}}}(\boldsymbol{y}|\boldsymbol{x})+\mathcal{F}^-(\boldsymbol{y}|\boldsymbol{x})\right), \tag{94}$$

which completes the proof. $\quad\square$

**Lemma A.4.** *For $0 \le a \le b \le 1$, let*

$$h_n(a,b) := \frac{n}{n+1}(b^{n+1}-a^{n+1}) - \frac{1}{2}(b^n-a^n)(b+a). \tag{95}$$

*We have*

$$h_n(a,b) \ge 0. \tag{96}$$

*Equivalently,*

$$\frac{1}{2}(b^n-a^n)(b+a) \le \frac{n}{n+1}(b^{n+1}-a^{n+1}). \tag{97}$$

*Proof.* Let us consider the function $h_n(a,v)$ for fixed $a$ as a function of $v$. Given that $h_n(a,a)=0$, it suffices to show that $\frac{\partial}{\partial v}h_n(a,v)\ge 0$ for all $0\le a\le v\le 1$. We have

$$\frac{\partial}{\partial v}h_n(a,v) = nv^n - \frac{n}{2}v^{n-1}(v+a) - \frac{1}{2}(v^n-a^n) \tag{98}$$

$$= \frac{1}{2}\left(nv^{n-1}(v-a)-(v^n-a^n)\right) \tag{99}$$

$$\ge 0, \tag{100}$$

completing the proof. $\quad\square$

**Lemma A.5.** *The following is an upper bound on $h_n(a,b)$ :*

$$h_n(a,b) \le \frac{n-1}{2}(b-a)^2. \tag{101}$$

*Proof.* We have

$$h_n(a,b) = \frac{n}{n+1}(b^{n+1}-a^{n+1}) - \frac{1}{2}(b^n-a^n)(b+a) \tag{102}$$

$$= (b-a)\left(\frac{n}{n+1}\sum_{i=0}^n b^i a^{n-i} - \frac{1}{2}(a+b)\sum_{i=0}^{n-1}b^i a^{n-1-i}\right) \tag{103}$$

$$= (b-a)\left(\left(\frac{n}{n+1}-\frac{1}{2}\right)(a^n+b^n)+\left(\frac{n}{n+1}-1\right)\sum_{i=1}^{n-1}a^i b^{n-i}\right) \tag{104}$$

$$\le (b-a)\left(\frac{n-1}{2(n+1)}(a^n+b^n)-\frac{n-1}{n+1}a^n\right) \tag{105}$$

$$= (b-a)\frac{n-1}{2(n+1)}(b^n-a^n) \tag{106}$$

$$\le (b-a)^2\frac{n(n-1)}{2(n+1)}b^{n-1} \tag{107}$$

$$\le \frac{n-1}{2}(b-a)^2, \tag{108}$$

completing the proof. $\quad\square$

*Proof of Theorem 5.3.* Note that

$$\mathcal{W}_r(\pi^{(n)}(\cdot|\boldsymbol{x}) \| \pi_{\text{ref}}(\cdot|\boldsymbol{x})) = \frac{1}{2} \sum_{\boldsymbol{y} \in \mathcal{Y}^*} \left( \mathcal{F}_{\pi_{\text{ref}}}(\boldsymbol{y}|\boldsymbol{x})^n - \mathcal{F}_{\pi_{\text{ref}}}^-(\boldsymbol{y}|\boldsymbol{x})^n \right) \left( \mathcal{F}_{\pi_{\text{ref}}}(\boldsymbol{y}|\boldsymbol{x}) + \mathcal{F}_{\pi_{\text{ref}}}^-(\boldsymbol{y}|\boldsymbol{x}) \right) \tag{109}$$

$$\leq \frac{n}{n+1} \sum_{\boldsymbol{y} \in \mathcal{Y}^*} \left( \mathcal{F}_{\pi_{\text{ref}}}(\boldsymbol{y}|\boldsymbol{x})^{n+1} - \mathcal{F}_{\pi_{\text{ref}}}^-(\boldsymbol{y}|\boldsymbol{x})^{n+1} \right) \tag{110}$$

$$= \frac{n}{n+1}, \tag{111}$$

where Equation (109) follows from Lemma 5.2 and Equation (110) follows from Lemma A.4. □

*Proof of Theorem 5.4.* We have

$$\frac{n}{n+1} - \mathcal{W}_r(\pi^{(n)}(\cdot|\boldsymbol{x}) \| \pi_{\text{ref}}(\cdot|\boldsymbol{x})) = \sum_{\boldsymbol{y} \in \mathcal{Y}^*} h_n(\mathcal{F}_{\pi_{\text{ref}}}^-(\boldsymbol{y}|\boldsymbol{x}), \mathcal{F}_{\pi_{\text{ref}}}(\boldsymbol{y}|\boldsymbol{x})) \tag{112}$$

$$\leq \frac{n-1}{2} \sum_{\boldsymbol{y} \in \mathcal{Y}^*} \left( \mathcal{F}_{\pi_{\text{ref}}}(\boldsymbol{y}|\boldsymbol{x}) - \mathcal{F}_{\pi_{\text{ref}}}^-(\boldsymbol{y}|\boldsymbol{x}) \right)^2 \tag{113}$$

$$= \frac{n-1}{2} \sum_{\boldsymbol{y} \in \mathcal{Y}^*} \left( \pi_{\text{ref}}(\boldsymbol{y}|\boldsymbol{x}) \right)^2 \tag{114}$$

$$= \frac{n-1}{2} e^{-H_2(\pi_{\text{ref}}|\boldsymbol{x})}, \tag{115}$$

where Equation (113) follows from Lemma A.5. □

*Proof of Theorem 5.6.* Recall Lemma A.4. Therefore,

$$G_{\mathcal{W}}^{(n)}(\boldsymbol{x}) \geq h_n(1 - \varepsilon_\infty, 1) \geq 0. \tag{116}$$

The proof is completed by noticing from Equation (95) that

$$h_n(1 - \varepsilon_\infty, 1) = \frac{n}{n+1} (1 - (1 - \varepsilon_\infty)^{n+1}) - (1 - (1 - \varepsilon_\infty)^n) \left( 1 - \frac{\varepsilon_\infty}{2} \right). \tag{117}$$

□

*Proof of Corollary 5.7.* As $n \to \infty$,

$$\frac{n}{n+1} (1 - (1 - \varepsilon_\infty)^{n+1}) - (1 - (1 - \varepsilon_\infty)^n) \left( 1 - \frac{\varepsilon_\infty}{2} \right) = \frac{\varepsilon_\infty}{2} (1 + o_n(1)), \tag{118}$$

which completes the proof. □

## A.4. Proofs of Section 6

*Proof of Lemma 6.1.* The proof is completed by observing that with probability $w_\Phi(\boldsymbol{x})$ a good outcome is obtained, and with probability $(1 - w_\Phi(\boldsymbol{x}))$ the trial is repeated. □

*Proof of Lemma 6.2.* First consider the KL divergence as follows:

$$D_{\mathrm{KL}}(\pi_\Phi(\cdot|\boldsymbol{x})\|\pi_{\mathrm{ref}}(\cdot|\boldsymbol{x})) = E_{\boldsymbol{y}\sim\pi_\Phi(\cdot|\boldsymbol{x})} \log \frac{\pi_\Phi(\boldsymbol{y}|\boldsymbol{x})}{\pi_{\mathrm{ref}}(\boldsymbol{y}|\boldsymbol{x})} \tag{119}$$

$$= E_{\boldsymbol{y}\sim\pi_\Phi(\cdot|\boldsymbol{x})} \log \frac{1}{w_\Phi(\boldsymbol{x})} \tag{120}$$

$$= \log \frac{1}{w_\Phi(\boldsymbol{x})}. \tag{121}$$

Next, let us consider the win rate:

$$\mathcal{W}_r(\pi_\Phi(\cdot|\boldsymbol{x})\|\pi_{\mathrm{ref}}(\cdot|\boldsymbol{x})) = (1 - w_\Phi(\boldsymbol{x})) \times 1 + w_\Phi(\boldsymbol{x}) \times \frac{1}{2} \tag{122}$$

$$= 1 - \frac{1}{2}w_\Phi(\boldsymbol{x}), \tag{123}$$

completing the proof. □

*Proof of Theorem 6.3.* Let $w_\Phi(\boldsymbol{x})$ be the probability of selecting a good sequence. Then, we note that

$$\frac{-\log(w_\Phi(\boldsymbol{x}))}{w_\Phi(\boldsymbol{x})} = \sum_{k=1}^{\infty} H_k(1 - w_\Phi(\boldsymbol{x}))^k,$$

where for $k \geq 1$ $H_k = \sum_{i=1}^{n} \frac{1}{i}$ is the $k$-th Harmonic number and $H_0 = 0$. Hence,

$$-\log(w_\Phi(\boldsymbol{x})) = \sum_{k=1}^{\infty} H_k(1 - w_\Phi(\boldsymbol{x}))^k w_\Phi(\boldsymbol{x}).$$

Let $M$ be the location of the first one (the number of trials). Then,

$$P(M = k) = (1 - w_\Phi(\boldsymbol{x}))^{k-1} w_\Phi(\boldsymbol{x}).$$

Hence, the following is an unbiased estimator of $-\log(w_\Phi(\boldsymbol{x}))$:

$$H_{M-1}.$$

□

## A.5. Proofs of Section 7

**Theorem A.6.** *For any $\boldsymbol{x}$ and a given $\pi_{ref}$, let the function $W_L(D_L)$ be implicitly defined for $\tau \geq 1$:*

$$W_L(\tau) = \frac{\tau}{\tau + 1} - \frac{\tau - 1}{2} e^{-H_2(\pi_{ref}|\boldsymbol{x})},$$

$$D_L(\tau) = \log(\tau) - \frac{\tau - 1}{\tau}.$$

*$W_L$ is a lower bound on the tradeoff curve of best-of-n.*

*Proof.* This is obtained by putting together Theorem 3.1 and Theorem 5.4. ☐

**Theorem A.7.** *For any $\boldsymbol{x}$ and a given $\pi_{ref}$, let the function $W_U(D_U)$ be implicitly defined for $\tau \geq 1$:*

$$W_U(\tau) = \frac{\tau}{\tau + 1},$$

$$D_U(\tau) = \log(\tau) - \frac{\tau - 1}{\tau} - 2\tau(\tau - 1)e^{-H_2(\pi_{ref}|\boldsymbol{x})}.$$

*$W_U$ is an upper bound on the tradeoff curve of best-of-n.*

*Proof.* This is obtained by combining Theorem 5.3 and Theorem 3.4. ☐

# B. KL divergence of blockwise best-of-n

While best-of-$n$ could be viewed as a sequence-level rejection sampling, most generative models perform decoding step by step. For example, language models perform generation token-by-token or diffusion models perform generation through several denoising steps. Best-of-$n$ could be extended to exert control at different levels of granularity. In particular, best-of-$n$ has been extended to provide control through blockwise decoding (Mudgal et al., 2024), where a block of length $B$ is decoded $n$ times and one with highest token-level reward is selected, and decoding is continued as such. Blockwise decoding could also be viewed as a simple form of tree search and also beam search. Our main result in this section characterizes the KL diveregnce of blockwise best-of-$n$ with respect to the reference policy.

For the purpose of this section, we assume that the fully decoded response $\boldsymbol{y} := (y_1, \ldots, y_T)$ consist of $T$ intermediate steps (which are tokens in the context of language model and denoising steps in the context of diffusion). For the purposes of this section, we focus on the presentation using generative language models. We let $y_T = \texttt{EOS}$ be a special token that determines the end of sequence. The autoregressive decoding could be further explained as follows. In the first decoding step, the model $\pi$ assigns a probability distribution over the first token given by $\pi(\cdot|\boldsymbol{x})$, and one token $y_1$ is drawn from this distribution. Next, the second token, $y_2$ is drawn from $\pi(\cdot|\boldsymbol{x}, y_1)$, and so on. In short, in each step a token is drawn from $\pi(\cdot|\boldsymbol{x}, y^{t-1})$ where $y^t := (y_1, \ldots, y_t)$. Note that through the chain rule, any language model could be equivalently expressed as a next-token predictor.

Blockwise best-of-$n$ decoding rule was recently proposed by Mudgal et al. (2024). Here, the decoder samples $n$ blocks of length $B$ tokens from the reference model and accepts one with highest reward. Formally, if the decoder has already decoded $y^t$ and the context is $\boldsymbol{x}$, then

$$z^B := \arg \max_{\{z^B_{(k)}\}_{k \in [n]}} r(x, y^t, z^B_{(k)}), \tag{124}$$

where $z^B_{(k)}$ is the $k$-th continuation of length $B$ drawn independently from the reference model:

$$\{z^B_{(k)}\}_{k \in [n]} \overset{i.i.d.}{\sim} \pi_{\text{ref}}(\cdot|\boldsymbol{x}, y^t), \tag{125}$$

and decoding continues until $\texttt{EOS}$ is reached. We call the resulting distribution $\pi_B^{(n)}$. Note that when $B \to \infty$, $\pi_B^{(n)}$ becomes the sequence level best-of-$n$ distribution, $\pi^{(n)}$.

Here, we assume we have access to a reward model can produce scalar values for $r(x, y^t)$ for any partially decoded sequence $y^t$, which extends the definition of a reward model to also score partial sequences in addition to fully decoded sequences. Mudgal et al. (2024) argue that for this extension to be meaningful the extended function should encode the *value function* for the sequence-level reward model. However, for the purposes of bounding the KL divergence, we don't care how the token-level reward function is obtained.

In the blockwise best-of-$n$ decoding, control is applied more frequently than the sequence-level best-of-$n$, and the blockwise decoding enables to effectively score an exponentially large number of fully decoded sequences (similarly to how beam search works). Thus, intuitively the effective $n$ would be exponentially larger and the KL divergence would be expected to grow linearly with the number of times control is applied, i.e., $|\boldsymbol{y}|/B$ where $|\boldsymbol{y}|$ is the length of the decoded sequence and $B$ is the block length. Next, we formalize this intuition by recalling Theorem B.1.

**Theorem B.1.** *Let a decoder, $\pi_B^{(n)}$, perform block-wise best-of-n with blocks of length $B$ steps. Further, let $\boldsymbol{y}$ be draw from this decoder in context $\boldsymbol{x}$ such that $|\boldsymbol{y}|$ is the length of $\boldsymbol{y}$, i.e., the total number of decoding steps. Then,*

$$D_{KL}(\pi_B^{(n)}(\cdot|\boldsymbol{x}) \| \pi_{ref}(\cdot|\boldsymbol{x})) \leq E_{\boldsymbol{y} \sim \pi_B^{(n)}(\cdot|\boldsymbol{x})} \left\lceil \frac{|\boldsymbol{y}|}{B} \right\rceil \widetilde{KL}_n,$$

*where $\lceil \cdot \rceil$ is the ceiling operator (smallest larger integer), and $\widetilde{KL}_n$ is defined in Equation (4).*

*Proof.* Let us consider all possible outcomes for decoding a block of length $B$. The outcomes either finish and we reach $\texttt{EOS}$ withing the block. We call this event $\mathcal{E} = 1$, or leads to a partially decoded sequence $\boldsymbol{z} = z^B$, and we call this outcome

$\mathcal{E} = 0$. In this case, we write $\boldsymbol{y} = (z^B, \boldsymbol{s})$.

$$D_{\mathrm{KL}}(\pi_B^{(n)}(\cdot|\boldsymbol{x})\|\pi_{\mathrm{ref}}(\cdot|\boldsymbol{x})) = E_{\boldsymbol{y}\sim\pi_B^{(n)}(\cdot|\boldsymbol{x})} \log \frac{\pi_B^{(n)}(\boldsymbol{y}|\boldsymbol{x})}{\pi_{\mathrm{ref}}(\boldsymbol{y}|\boldsymbol{x})} \tag{126}$$

$$= E_{\boldsymbol{y}\sim\pi_B^{(n)}(\cdot|\boldsymbol{x})}\left\{\mathbf{1}(\mathcal{E}=1)\log\frac{\pi_B^{(n)}(\boldsymbol{y}|\boldsymbol{x})}{\pi_{\mathrm{ref}}(\boldsymbol{y}|\boldsymbol{x})}\right\} + E_{\boldsymbol{y}\sim\pi_B^{(n)}(\cdot|\boldsymbol{x})}\left\{\mathbf{1}(\mathcal{E}=0)\log\frac{\pi_B^{(n)}(\boldsymbol{y}|\boldsymbol{x})}{\pi_{\mathrm{ref}}(\boldsymbol{y}|\boldsymbol{x})}\right\} \tag{127}$$

$$= E_{\boldsymbol{y}\sim\pi_B^{(n)}(\cdot|\boldsymbol{x})}\left\{\mathbf{1}(\mathcal{E}=1)\log\frac{\pi_B^{(n)}(\boldsymbol{y}|\boldsymbol{x})}{\pi_{\mathrm{ref}}(\boldsymbol{y}|\boldsymbol{x})}\right\}$$
$$+ E_{z^B\sim\pi_B^{(n)}(\cdot|\boldsymbol{x})}E_{\boldsymbol{s}\sim\pi_B^{(n)}(\cdot|\boldsymbol{x},z^B)}\left\{\mathbf{1}(\mathcal{E}=0)\left(\log\frac{\pi_B^{(n)}(z^B|\boldsymbol{x})}{\pi_{\mathrm{ref}}(z^B|\boldsymbol{x})} + \log\frac{\pi_B^{(n)}(\boldsymbol{s}|\boldsymbol{x},z^B)}{\pi_{\mathrm{ref}}(\boldsymbol{s}|\boldsymbol{x},z^B)}\right)\right\} \tag{128}$$

$$= E_{\boldsymbol{y}\sim\pi_B^{(n)}(\cdot|\boldsymbol{x})}\left\{\mathbf{1}(\mathcal{E}=1)\log\frac{\pi_B^{(n)}(\boldsymbol{y}|\boldsymbol{x})}{\pi_{\mathrm{ref}}(\boldsymbol{y}|\boldsymbol{x})}\right\} + E_{z^B\sim\pi_B^{(n)}(\cdot|\boldsymbol{x})}\left\{\mathbf{1}(\mathcal{E}=0)\log\frac{\pi_B^{(n)}(z^B|\boldsymbol{x})}{\pi_{\mathrm{ref}}(z^B|\boldsymbol{x})}\right\}$$
$$+ E_{z^B\sim\pi_B^{(n)}(\cdot|\boldsymbol{x})}E_{\boldsymbol{s}\sim\pi_B^{(n)}(\cdot|\boldsymbol{x},z^B)}\left\{\mathbf{1}(\mathcal{E}=0)\log\frac{\pi_B^{(n)}(\boldsymbol{s}|\boldsymbol{x},z^B)}{\pi_{\mathrm{ref}}(\boldsymbol{s}|\boldsymbol{x},z^B)}\right\} \tag{129}$$

$$\leq \widetilde{\mathrm{KL}}_n + E_{z^B\sim\pi_B^{(n)}(\cdot|\boldsymbol{x})}\left\{\mathbf{1}(\mathcal{E}=0)D_{\mathrm{KL}}(\pi_B^{(n)}(\cdot|\boldsymbol{x},z^B)\|\pi_{\mathrm{ref}}(\cdot|\boldsymbol{x},z^B))\right\} \tag{130}$$

$$\leq \widetilde{\mathrm{KL}}_n + E_{z^B\sim\pi_B^{(n)}(\cdot|\boldsymbol{x})}\mathbf{1}(\mathcal{E}=0)E_{\boldsymbol{s}\sim\pi_B^{(n)}(\cdot|\boldsymbol{x},z^B)}\left\lceil\frac{|\boldsymbol{s}|}{B}\right\rceil\widetilde{\mathrm{KL}}_n \tag{131}$$

$$= E_{\boldsymbol{y}\sim\pi_B^{(n)}(\cdot|\boldsymbol{x})}\left\lceil\frac{|\boldsymbol{y}|}{B}\right\rceil\widetilde{\mathrm{KL}}_n, \tag{132}$$

where Equation (131) follows from recursively applying the same procedure to the subsequent blocks. $\qquad\square$

This theorem immediately suggests that $\lceil\frac{|\boldsymbol{y}|}{B}\rceil\widetilde{\mathrm{KL}}_n$ could be used as an estimator for the KL divergence of block-wise best-of-$n$, and in expectation the estimator provides an upper bound on the KL divergence of blockwise best-of-$n$ and the reference model. Sequence-level best-of-$n$ could be viewed as blockwise best-of-$n$ with $B \to \infty$, and Theorem B.1 simplifies to Theorem 3.1 in this asymptotic regime.

# C. Experiments

## C.1. Details of experiments on Alpaca dataset

We select the following four prompts from the Alpaca dataset (Taori et al., 2023).

P1 Transform the following sentence into a yes/no question. It is going to rain tomorrow.

P2 Describe the function of a computer motherboard.

P3 What is the capital of France?

P4 Give three tips for staying healthy.

We plot results based on Gemma 2 9B parameter instruction tuned model (Gemma et al., 2024) with temperature one. We further modify the prompts as per the instructions in `https://ai.google.dev/gemma/docs/formatting`. We use log-likelihood of the reference model as the reward. To get the better upper bound we use Corollary 4.2, where we bound $\varepsilon_\infty$ using 100 samples.

## C.2. Computation of tilted min/max

We compute the tilted average of the estimator defined by (Li et al., 2023, Equation (2)):

$$\frac{1}{t} \log E_{\boldsymbol{y} \sim \pi^{(n)}(\cdot|\boldsymbol{x})} \left[ e^{t d_n(\varepsilon_n)} \right]. \tag{133}$$

Note that Equation (133) recovers $\min_{\boldsymbol{y} \sim \pi^{(n)}(\cdot|\boldsymbol{x})} [d_n(\varepsilon_n)]$ for $t \to -\infty$, and $\max_{\boldsymbol{y} \sim \pi^{(n)}(\cdot|\boldsymbol{x})} [d_n(\varepsilon_n)]$ for $t \to \infty$ (Li et al., 2023). To capture the variance, we call the value for $t = -1$ the *tilted min*, and the value for $t = 1$ the *tilted max*. The tilted min/max capture the low/high quantiles of the value of the estimator and their difference portrays the deviation from the mean.

## C.3. Experiments with machine translation prompts

In this section, we demonstrate the value of the new estimator using machine translation prompts.

P1 Translate the next sentences to German. I want to buy bread.

P2 Translate the next sentences to French. I want to buy bread.

P3 Translate the next sentences to German. A simple and effective method for the inference-time alignment and scaling test-time compute of generative models is the best-of-$n$ policy, where $n$ samples are drawn from a reference policy, ranked based on a reward function, and the highest ranking one is selected. A commonly used analytical expression in the literature claims that the KL divergence between the best-of-$n$ policy and the reference policy is equal to $\log(n) - (n-1)/n$. We disprove the validity of this claim, and show that it is an upper bound on the actual KL divergence. We also explore the tightness of this upper bound in different regimes, and propose a new estimator for the KL divergence and empirically show that it provides a tight approximation. We also show that the win rate of the best-of-$n$ policy against the reference policy is upper bounded by $n/(n+1)$ and derive bounds on the tightness of this characterization. We conclude with analyzing the tradeoffs between win rate and KL divergence of the best-of-$n$ alignment policy, which demonstrate that very good tradeoffs are achievable with $n < 1000$.

P4 Translate the next sentences to French. A simple and effective method for the inference-time alignment and scaling test-time compute of generative models is the best-of-$n$ policy, where $n$ samples are drawn from a reference policy, ranked based on a reward function, and the highest ranking one is selected. A commonly used analytical expression in the literature claims that the KL divergence between the best-of-$n$ policy and the reference policy is equal to $\log(n) - (n-1)/n$. We disprove the validity of this claim, and show that it is an upper bound on the actual KL divergence. We also explore the tightness of this upper bound in different regimes, and propose a new estimator for the KL divergence and empirically show that it provides a tight approximation. We also show that the win rate of the best-of-$n$ policy against the reference policy is upper bounded by $n/(n+1)$ and derive bounds on the tightness of this characterization. We conclude with analyzing the tradeoffs between win rate and KL divergence of the best-of-$n$ alignment policy, which demonstrate that very good tradeoffs are achievable with $n < 1000$.

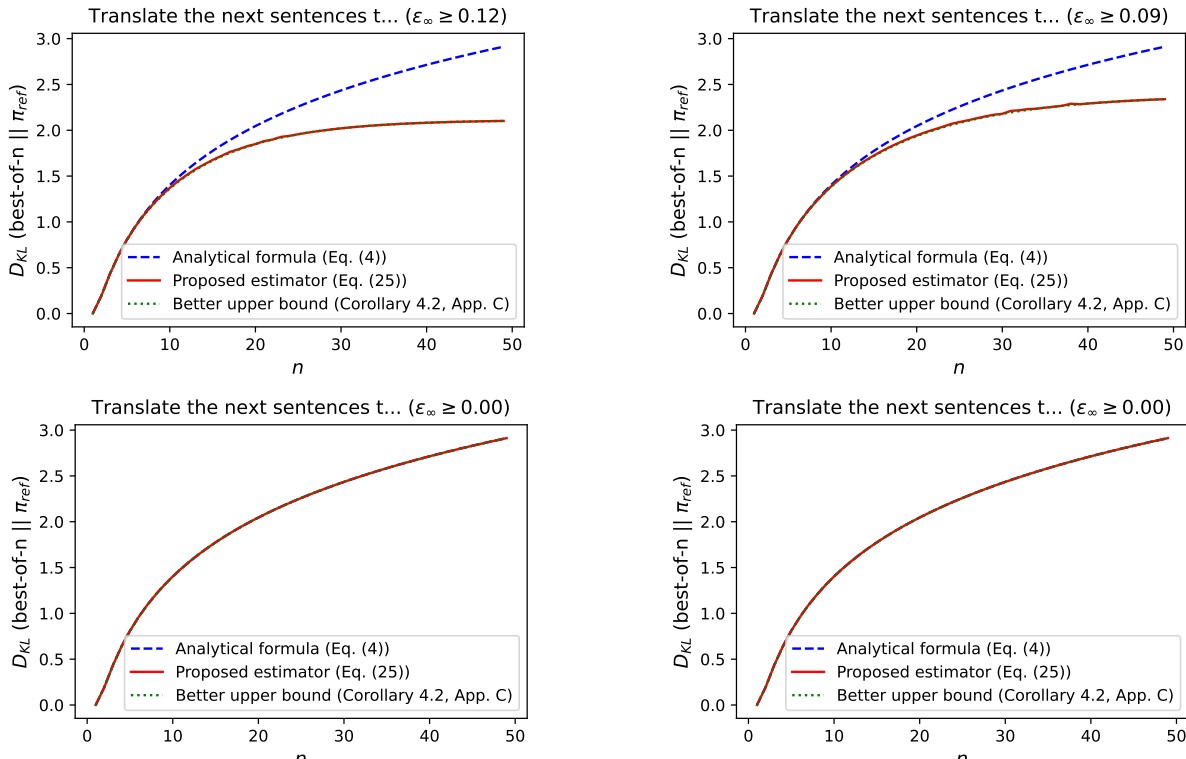

*Figure 10.* The analytical formula $(\log(n) - (n-1)/n)$, Equation (4), better upper bound (Corollary 4.2, Appendix C), the proposed estimator, Equation (25), and the exact KL divergence, for four machine translation examples using Gemma 9B IT model (Gemma et al., 2024) with reward the log-likelihood of response under the reference model.

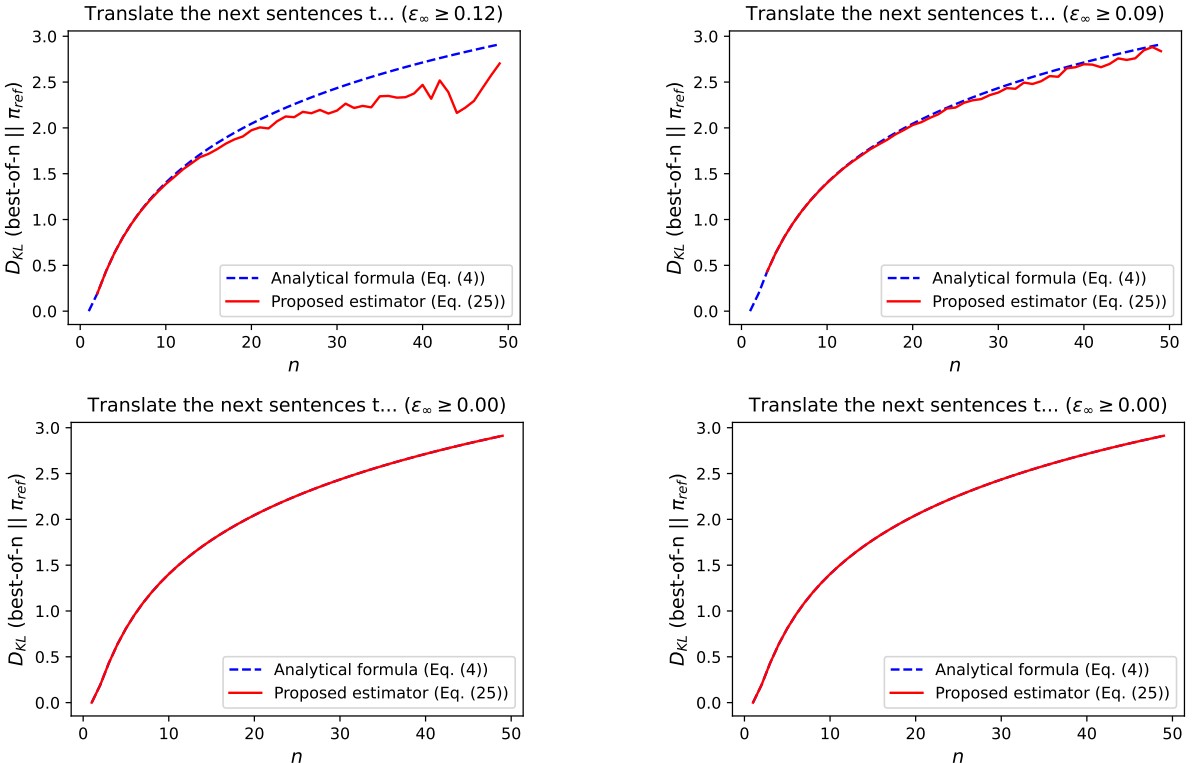

*Figure 11.* The analytical formula $(\log(n) - (n-1)/n)$, Equation (4), better upper bound (Corollary 4.2, Appendix C), the proposed estimator, Equation (25), and the exact KL divergence, for four machine translation examples using Gemma 9B IT model (Gemma et al., 2024) with reward the negative length of response under the reference model.

