# OpenReview forum: "Theoretical guarantees on the best-of-n alignment policy"
_ICML.cc/2025/Conference — ICML 2025 poster_

### Official Review · Reviewer_nvEQ · 2025-03-10

**Overall Recommendation:** 3

**Summary:**

This paper provides a theoretical analysis of the best-of-$n$ policy $\pi^{(n)}$, which is a simple inference-time method for aligning language models where $n$ samples are drawn from a reference policy $\pi_{ref}$, the highest ranking one based on a reward function is selected.

The authors first disprove a commonly used analytical expression for $KL (\pi^{(n)} || \pi_{ref})$. They demonstrate that the formula $\widetilde{KL}\_n := \log(n) - \frac{n-1}{n}$, which is widely cited in the literature, is actually just an upper bound on the true KL divergence. They theoretically characterize when this bound is tight and when it can be arbitrarily loose.
Additionally, the paper develops a new estimator for $KL (\pi^{(n)} || \pi_{ref})$ that more accurately tracks the true value, as demonstrated by experiments.
The authors also analyze the win rate of $\pi^{(n)}$ against $\pi_{ref}$. This analysis shows that the win rate is upper bounded by $\frac{n}{n+1}$ and characterizes when this bound is tight.

The authors also compare best-of-$n$ with another rejection sampling approach called rewind-and-repeat, ultimately showing the superiority of best-of-$n$ in terms of win rate vs. KL divergence tradeoffs.
The paper concludes that the actual win rate vs. KL divergence tradeoffs for best-of-$n$ are better than what has been reported in the literature when using the incorrect formula, and that very good tradeoffs are achievable with $n < 1000$.

**Claims And Evidence:**

Yes

**Essential References Not Discussed:**

No

**Experimental Designs Or Analyses:**

I reviewed all experiments mentioned in the main text. My concern is with the experiment in Figure 4, where the authors test language models on benchmarks. The reward is set as the log-likelihood of the reference model, which already correlates with the policies, thus reducing generality. I suggest the authors provide more explanation about this choice, justify this approach, or discuss a more general reward function.

**Methods And Evaluation Criteria:**

Yes

**Other Comments Or Suggestions:**

No

**Other Strengths And Weaknesses:**

### Strengths
- **Theoretical guarantees:** The paper provides mathematical analysis, establishing bounds on both KL divergence and win rates. It corrects a misunderstanding in the literature by showing the formula $\widetilde{KL}\_n := \log(n) - \frac{n-1}{n}$ is only an upper bound for $KL (\pi^{(n)} || \pi_{ref})$.
- **Practical estimator:** The proposed KL divergence estimator has practical value for researchers evaluating alignment methods, as it more accurately captures the true KL divergence, as demonstrated by experiments.
- **Comparative analysis:** The comparison with rewind-and-repeat provides context on why best-of-$n$ performs well in practice through the lens of tradeoffs between KL divergence and win rates.

### Weaknesses
- **Limited empirical validation:** While the paper includes examples, it lacks extensive empirical validation on real language models and datasets. More comprehensive experiments would strengthen the claims. Additionally, in the experiment in Figure 4, the reward is highly correlated with the reference policy, which represents a constrained case.

**Questions For Authors:**

Regarding the empirical validation on benchmarks, what will the results be if the reward model is no longer the log likelihood of $\pi_{ref}$?

**Relation To Broader Scientific Literature:**

This paper is significant to the field of language model alignment for several reasons:

- It corrects a fundamental misunderstanding about the KL divergence of best-of-$n$. It also provides theoretical justification for why best-of-$n$ performs so well empirically, showing that its win rate vs. KL divergence tradeoffs are actually better than previously thought.
- It offers theoretical bounds and a more accurate estimator for KL divergence that can be used to better evaluate alignment methods. These results can give hints on choosing the best $n$ in practice.

**Theoretical Claims:**

No, but they seems to be correct by intuition.

---

> ### Author Rebuttal · Authors · 2025-04-01
>
> Thanks for your insightful review and encouraging comments.
>
>
> > My concern is with the experiment in Figure 4, where the authors test language models on benchmarks. The reward is set as the log-likelihood of the reference model, which already correlates with the policies, thus reducing generality. I suggest the authors provide more explanation about this choice, justify this approach, or discuss a more general reward function.
>
> We have now experimented with other tasks and other rewards (e.g., generation length) to showcase that this is not just an issue that pertains to the log-likelihood reward. See https://anonymous.4open.science/r/bon.
> We would like to reiterate that anytime there is a non-trivial chance of collision between the outcomes in a set of size $n$, the analytical formula for the KL divergence overestimates the true KL.

---

> > ### Comment · Reviewer_nvEQ · 2025-04-02
> >
> > Thank you for the additional experiment results -- they look good to me. I'll keep my positive score.

---

### Official Review · Reviewer_3Tjm · 2025-03-13

**Overall Recommendation:** 3

**Summary:**

This paper discusses the best-of-n policy, a method used for inference-time alignment of generative models. The key idea is to draw n samples from a reference policy, rank them based on a reward function, and select the highest-ranking sample. The passage critiques a commonly used analytical expression in the literature, which claims that the KL divergence between the best-of-n policy and the reference policy is equal to log(n)−(n−1)/n. The authors disprove this claim, showing that it is instead an upper bound on the actual KL divergence. They also explore the tightness of this bound and propose a new estimator for the KL divergence that provides a tighter approximation.
Additionally, the authors analyze the win rate of the best-of-n policy against the reference policy, showing that it is upper bounded by n/(n+1) . They derive bounds on the tightness of this characterization and conclude by analyzing the tradeoffs between win rate and KL divergence. Their findings suggest that very good tradeoffs can be achieved with n<1000

**Claims And Evidence:**

The claims made in the submission supported by clear and convincing evidence. Assumption 2.2: It is posited that the output space of the language model is finite. This assumption simplifies the theoretical analysis; however, in practical applications, the output space of generative models is typically infinite (as in natural language generation tasks), which may limit the universality of the theory.

**Essential References Not Discussed:**

Most of the essential references have already been cited within this paper.

**Experimental Designs Or Analyses:**

The experiment did not involve validation with real generative models, resulting in uncertainty regarding the practical application effectiveness of the theoretical results. In particular, the output space of real generative models is typically large and complex, necessitating further verification of the applicability of the theoretical results in these scenarios.

**Methods And Evaluation Criteria:**

The proposed methods and evaluation criteria make good sense for the problem or application at hand.

**Other Comments Or Suggestions:**

I do not have any other comments or suggestions.

**Other Strengths And Weaknesses:**

Strengths:
1. This paper not only analyzes the KL divergence but also delves into the win rate of the best-of-n strategy, proving an upper bound of the win rate to be n/(n+1). Furthermore, the authors derive a trade-off relationship between the win rate and KL divergence, demonstrating how adjusting n can achieve a better balance between performance and model alignment in various scenarios.
2. This paper introduces a novel estimator for KL divergence and validates its effectiveness through numerical experiments. This estimator provides a more accurate reflection of the KL divergence between the best-of-n strategy and the reference strategy.

Weaknesses:
1. This paper does not provide detailed guidance on the specific implementation and optimization of the best-of-n strategy in practical applications.

**Questions For Authors:**

I do not have any important questions for the authors.

**Relation To Broader Scientific Literature:**

This paper refutes the commonly used expression for KL divergence in the literature, demonstrating that it merely serves as an upper bound to the actual KL divergence. Furthermore, a novel estimator for KL divergence is proposed.

**Theoretical Claims:**

I have examined all the theoretical proofs to ensure their accuracy.

---

> ### Author Rebuttal · Authors · 2025-04-01
>
> Thanks for your insightful review and encouraging comments.
>
> > This assumption simplifies the theoretical analysis; however, in practical applications, the output space of generative models is typically infinite (as in natural language generation tasks), which may limit the universality of the theory.
>
> We would like to mention that (1) in practice, language models have a finite output tokens and will eventually stop hence practically the number of possible generated sequences is finite (even though it could be very large), which is covered by our theory; (2) this assumption could be relaxed using limit arguments, and we are happy to include a relaxation of it; (3) this result has already been extended to models with potentially continuous outputs (which would not even be countable) by Mroueh (2024) (subsequent to our work) so the extension beyond what is proved here is certainly possible.
>
> Mroueh, Youssef. "Information theoretic guarantees for policy alignment in large language models." arXiv preprint arXiv:2406.05883 (2024).
>
>
> > The experiment did not involve validation with real generative models, resulting in uncertainty regarding the practical application effectiveness of the theoretical results. In particular, the output space of real generative models is typically large and complex, necessitating further verification of the applicability of the theoretical results in these scenarios.
>
> We would like to mention that we already showed a certain case with prompts from AlpacaEval dataset and Gemma2 IT 9B in Figure 4. We have now expanded the scope of experiments with more tasks and rewards in https://anonymous.4open.science/r/bon
>
> > This paper does not provide detailed guidance on the specific implementation and optimization of the best-of-n strategy in practical applications.
>
> While we agree with the reviewer that we do not provide an optimization of best-of-n, our results imply that with n~100-1000, win rate against base model will be already saturated without the need for excessively large n which also implies that KL divergence of ~5 is enough to reach a good policy. This implies that RLHF practitioners can aim to keep the KL divergence of their aligned models <10 and achieve good policies, which also significantly helps mitigate reward overoptimization.

---

### Official Review · Reviewer_jbod · 2025-03-14

**Overall Recommendation:** 3

**Summary:**

This paper revisits the best-of-n alignment policy: given n samples from a reference language model, pick the sample that scores highest under a reward (alignment) function. A long-used formula in prior papers, $D_{kl}=log(n)-(n-1)/n$, has been cited as the KL divergence of the best-of-n policy from the reference. The authors show that although this expression frequently appears, it is only an upper bound—not an exact value—in many realistic scenarios. They derive exact or tighter bounds, provide a new practical KL estimator, and show that in some regimes (particularly if a small number of high-probability completions exist under the reference), the conventional formula can substantially overestimate the real KL drift.

In addition to these theoretical clarifications, they also: (1) Analyze blockwise best-of-n (where re-ranking occurs multiple times during generation). (2) Compare best-of-n to “rewind-and-repeat,” another rejection-sampling-like policy. (3) Show how their refined analysis can help practitioners accurately measure or cap KL divergence in alignment pipelines—important for settings where controlling distribution shift is crucial (e.g., compliance, RLHF with KL constraints).

**Claims And Evidence:**

The paper’s key claims are relatively well substantiated by proofs and by numeric examples carefully chosen to highlight the regimes where the old formula fails.

**Essential References Not Discussed:**

The authors already cite many highly relevant alignment and preference-based optimization works (including those that have used or mentioned the log(n) formula for best-of-n). No glaring omission is evident.

Overall, the bibliography covers standard RLHF, preference optimization, and controlled decoding references.

**Experimental Designs Or Analyses:**

The chosen synthetic examples clearly demonstrate the paper’s main theoretical points.The real prompts confirm that skewed probability distributions are common in real queries.

**Potential Weaknesses**:
The real data experiments are limited and do not show large-scale or broad tasks.
The variance of the proposed KL estimator in practical, large n sampling is not deeply explored.

**Methods And Evaluation Criteria:**

The real data experiments are limited and do not show large-scale or broad tasks.
The variance of the proposed KL estimator in practical, large n sampling is not deeply explored.

**Other Comments Or Suggestions:**

- (1) While it is likely that in many real tasks with big n and small probabilities per token, the difference is small, showing this in a broader study would underscore the conditions under which the new analysis provides a “significant” improvement.
- (2)  Because the paper is about “how far the policy can deviate,” it might be worth an explicit mention that tighter estimates can help compliance or risk-management teams ensure that the model does not drift too far from a safe baseline.
- (3) The authors’ theoretical perspective might help design an adaptive best-of-n procedure that halts once an approximate KL threshold is reached, or once a certain minimum reward is achieved.

**Other Strengths And Weaknesses:**

**Strengths**
- Clarification of a widely cited formula: The old rule was occasionally treated as exact, which could be misleading in some alignment or compliance scenarios.
- Practical new estimator: Offers a run-time method to measure “how much drift best-of-n actually used,” potentially letting practitioners adapt n on the fly or confirm they have not exceeded a KL budget.
- Extensions: The coverage of blockwise decoding and comparisons to alternative rejection sampling (rewind-and-repeat) broadens applicability.

**Weaknesses**
- Empirical scope: The paper mostly uses small or contrived examples. Large-scale, high-n tasks in more varied domains could strengthen claims about real-world use.
- Variance analysis: The new estimator’s variance or confidence intervals in complex distributions is left for future work.

**Questions For Authors:**

**Estimator Variance**
- How large can the variance of the proposed KL estimator get in real usage (especially if $\epsilon_n$  is extremely small)?
- Would you recommend running it repeatedly and averaging over many sequences, or do you envision an online, per-sample approach?

**Blockwise vs. Full-Sequence**
- Have you observed in practice (beyond the toy examples) that blockwise best-of-n greatly outperforms single-step best-of-n in reward–KL tradeoffs, or do you expect diminishing returns?

**Real-World Deployments**
- Can you give a concrete scenario (e.g. in compliance or enterprise alignment) where the old formula’s overestimation of KL meaningfully hampered performance or forced overly conservative constraints?
A real case study would clarify the direct practical payoff.

**Infinite/Very Large Vocabularies**
- Your proofs rely on finite support or a finite set of “possible outcomes” in each context. For large-vocabulary LMs, do you see a direct extension or an approximate argument? Would it still hold to treat large but finite vocab sizes similarly?

**Relation To Broader Scientific Literature:**

- **Alignment & KL Regularization**: The paper ties into RLHF and reward-based alignment (Christiano et al., Ouyang et al., etc.). Many works track or constrain $KL(\pi||\pi_{ref} )$ as a measure of “distribution drift.”

- **Controlled Decoding**: Ties well to prior re-ranking or decoding approaches like beam search, blockwise best-of-n, or other re-ranking methods.

- **Relation to Rejection Sampling**: They connect best-of-n and “rewind-and-repeat” as forms of rejection sampling, referencing literature on approximate generation, safe decoding, or iterative expansions.

- **Practical Gains**: The new results clarify that in a “$\delta$-bound” setting (where the reference rarely repeats outcomes) the old formula is indeed tight, but in the presence of a few large-prob outcomes, the real KL can be drastically smaller.

**Theoretical Claims:**

The main theorems (e.g. Theorem 3.1, Theorem 3.4, Theorem 5.3) use standard tools from information theory (KL divergence definitions, integrals, combinatorial arguments) and appear consistent. The proofs in the appendices logically match the statements in the main text.

**Potential concerns**: The theorems assume finite support or at least that the model probabilities for outcomes can be well-defined in a somewhat discrete sense. Real LMs often have large vocabularies, but the authors argue that for bounding or approximate computation, this finite-support assumption can be relaxed in practice.

---

> ### Author Rebuttal · Authors · 2025-04-01
>
> Thanks for your insightful review and encouraging comments
>
> > The real data experiments are limited and do not show large-scale or broad tasks.
>
> The goal of the experiments was to show cases where the could be a gap between the analytical formula and the true KL. We have now experimented with other prompts for machine translation and other rewards, namely output length, and still show that the KL estimates could be loose in more tasks. See https://anonymous.4open.science/r/bon
>
> > The variance of the proposed KL estimator in practical, large n sampling not deeply explored. Would you recommend running it repeatedly and averaging over many sequences, or do you envision an online, per-sample approach?
>
> Thanks for this very important question. The proposed estimator is upper bounded by the analytical formula, and hence by log(n). It is also non-negative and lower bounded by 0. Hence, a crude upper bound on its standard deviation is $\log(n)$. Thus, if the estimator is averaged out over $M = O(\log(n) \log\frac{1}{\delta})$ responses the standard deviation could be driven down to below $\delta$. Given that we are generally interested in $n < 1000$, the dependence on $n$ is manageable. Having said that, given each of the M batches has $n$ iid samples (total of $M \times n$ iid samples), we can use a bootstrapped estimator and should be able to remove the dependence on $\log(n)$. We will think more about the variance estimation and include it in the next iteration.
>
> We would like to also mention that the variance of the estimator in Figure 2 is exactly zero given all outcomes are the same likelihood. We have now also computed and plotted the standard deviation of the estimator in Figure 3 in the new PDF.  See https://anonymous.4open.science/r/bon.
>
> > The theorems assume finite support or at least that the model probabilities for outcomes can be well-defined in a somewhat discrete sense. Real LMs often have large vocabularies ...
>
> The finite support assumption could be relaxed in practice by limit arguments given that PMF is bounded, and we will include this relaxation in the next version. We would like to also mention that subsequent to our work, Mroueh (2024) in fact has proved a variant of Theorem 3.1 under much weaker assumptions that even applies to continuous distributions such as diffusion process.
>
> Mroueh, Youssef. "Information theoretic guarantees for policy alignment in large language models." arXiv preprint arXiv:2406.05883 (2024).
>
> > Showing this in a broader study would underscore the conditions under which the new analysis provides a “significant” improvement. A concrete scenario (e.g. in compliance or enterprise alignment) where the old formula’s overestimation of KL meaningfully hampered performance?
>
> We would like to emphasize that the message of this paper is not just about providing a new estimator for KL divergence. We also prove that the existing formula is an upper bound and hence the works that use it still give guarantees on the win rate vs KL divergence of best-of-n.
>
> > Because the paper is about “how far the policy can deviate,” it might be worth an explicit mention that tighter estimates can help compliance or risk-management teams ensure that the model does not drift too far from a safe baseline.
>
> Thanks for the suggestion. Will explicitly mention it.
>
> > The authors’ theoretical perspective might help design an adaptive best-of-n procedure that halts once an approximate KL threshold is reached, or once a certain minimum reward is achieved.
>
> Best-of-infinity with halting when a certain threshold on reward is reached is akin to rewind-and-repeat which we analyze in Section 6, where we show that the resulting tradeoff between win rate and KL is less favorable compared to best-of-n. However, the resulting tradeoff between win rate and cost (number of decoding trials) is more favorable than best-of-n, which is important in test-time scaling. Hence, it remains to be seen how to best design a method that achieves Pareto optimal tradeoffs between compute, reward, and KL divergence (which captures preservation of capabilities other than what is captured by reward), which might involve combining the rewind-and-repeat with best-of-n for some finite n as the reviewer suggests. We will add a discussion to this end in the concluding remarks as an area for future work.
>
> > Have you observed in practice (beyond the toy examples) that blockwise best-of-n greatly outperforms single-step best-of-n in reward–KL tradeoffs, or do you expect diminishing returns?
>
> Blockwise best-of-n has been studied comprehensively by (Mudgal et al., 2024). While the reward vs KL tradeoffs are generally not better than best-of-n (due to the fact that best-of-n is already almost optimal), blockwise best-of-n generally allows to achieve similar reward vs KL tradeoffs with ~10x smaller n, which means a 10x reduction in test-time compute.
>
> Mudgal, Sidharth, et al. "Controlled Decoding from Language Models." ICML, 2024.

---

> > ### Comment · Reviewer_jbod · 2025-04-04
> >
> > Thank you for the detailed and thoughtful rebuttal, as well as for addressing the concerns I raised in the review. I appreciate the additional information and clarifications you have provided, and I acknowledge the efforts to improve the paper based on the feedback.
> >
> > ---
> > 1. *Real Data Experiments*:
> > I’m glad to see that you’ve conducted additional experiments with various tasks, such as machine translation and output length as a reward. While I appreciate these further experiments, I still believe that more large-scale, varied real-world tasks would strengthen the empirical validation of your claims, particularly for more complex scenarios. It would be beneficial to see how your findings generalize across a wider range of tasks, especially with larger models.
> > ---
> > 2. *KL Estimator Variance*:
> > Your explanation of the variance of the KL estimator and the use of bootstrapping is insightful. I appreciate the inclusion of standard deviation plots in the new version, which help clarify the behavior of the estimator in practice. However, I still think that further exploration of the variance, particularly in high n scenarios, would provide a clearer understanding of the estimator's reliability in real-world settings. As mentioned, this could be explored in future iterations to ensure the estimator’s stability across different distributions.
> > ---
> > 3. *Finite Support Assumption*:
> > Thank you for the clarification regarding the finite support assumption and the reference to Mroueh’s (2024) work. It’s reassuring to know that there is progress on relaxing these assumptions for continuous distributions, and I look forward to seeing the extension of your results in that direction in the next version. This would further bolster the applicability of your work to large-vocabulary language models.
> > ---
> > 4. *Concrete Scenario and Practical Impact*:
> > I appreciate your emphasis on the theoretical contribution, which shows that the old formula is an upper bound. I also welcome the decision to explicitly mention the impact of tighter estimates for compliance and risk-management teams, which would make the practical value of your work even more apparent. As you mention, understanding where the old formula’s overestimation hampers performance in real cases is an important next step, and I look forward to seeing such discussions in future versions of the paper.
> > ---
> > 5. *Adaptive Best-of-n Procedure*:
> > I find the comparison to rewind-and-repeat insightful, and I agree that combining both methods for a Pareto-optimal tradeoff between compute, reward, and KL divergence could be a promising direction for future work. I look forward to the addition of this discussion in the concluding remarks, as it presents a valuable avenue for enhancing best-of-n’s applicability in real-world use cases.
> > ---
> > 6. *Blockwise Best-of-n*:
> >  Thank you for the clarification regarding blockwise best-of-n and the work by Mudgal et al. (2024). The reduction in test-time compute is indeed a compelling reason for using blockwise best-of-n, especially in large-scale scenarios. Although, as you note, the tradeoffs may not always be better than single-step best-of-n, it seems that the approach could still yield practical benefits in terms of compute efficiency.
> > ---
> > Overall, I appreciate the detailed response and the additional insights provided. The clarifications regarding the variance of the estimator, blockwise best-of-n, and the relaxation of finite support assumptions strengthen the paper’s contributions. While I still believe that more extensive real-world experiments and further exploration of variance would be valuable, I recognize the theoretical and practical significance of your work, and I maintain my overall score.

---

> > > ### Author Response · Authors · 2025-04-08
> > >
> > > We greatly appreciate the reviewer taking the time to engage with our paper and responses and providing an additional round of feedback. We are fully in agreement with your assessment and will provide individual responses below.
> > >
> > >
> > > > Real Data Experiments
> > >
> > > We agree with the reviewer that more large-scale experiments would strengthen the findings, especially for practitioners. We plan to perform additional experiments on all prompts from Anthropic Helpfulness, Harmlessness, and Reddit text summarization to broaden the scope of the empirical study in the next version of this paper. Due to the limited time, we couldn't get to this during the rebuttal period. Please let us know if you have any other suggestions.
> > >
> > > > KL Estimator Variance
> > >
> > > We agree that better understanding the variance of the estimator is important. As promised we will include theoretical bounds on the variance of the KL estimator in the next version of the paper and will explore understanding it better to the extent possible.
> > >
> > > > Finite Support Assumption
> > >
> > > As promised, we will include this relaxation in the next version of the paper.
> > >
> > > > Concrete Scenario and Practical Impact
> > >
> > > As promised, we will include these discussion points in the next version of the paper.
> > >
> > > > Adaptive Best-of-n Procedure
> > >
> > > Thanks again for this suggestion. We will include these discussion points in the next version of the paper.
> > >
> > > > Overall, I appreciate the detailed response and the additional insights provided. The clarifications regarding the variance of the estimator, blockwise best-of-n, and the relaxation of finite support assumptions strengthen the paper’s contributions. While I still believe that more extensive real-world experiments and further exploration of variance would be valuable, I recognize the theoretical and practical significance of your work, and I maintain my overall score.
> > >
> > > Thanks again for your insightful feedback, and for engaging with us in multiple rounds of discussions. We are committed to addressing these shortcomings to the full extent in the next version of the paper.

---

### Official Review · Reviewer_vBEz · 2025-03-14

**Overall Recommendation:** 4

**Summary:**

The paper provides theoretical analyses for the widely used Best-of-N (BoN) policy, especially focusing on the KL divergence from the reference model and its win-rate. They first point out that the conventionally used formula for the KL divergence does not hold and actually gives only an upper bound. They evaluate the gap between the true KL divergence and the upper bound, showing the non-triviality of the gap under when N is sufficiently large (Section 3). They also propose an alternative estimator for the KL divergence, which is still a conjecture but experimental results indicate the plausibility of the conjecture (Section 4). Next, they analyze the win-rate of the BoN against the reference model, and provide an upper bound and its tightness evaluation (Section 5). Meanwhile, they also analyze the rewind-and-repeat model $\pi_{\Phi}$ with a threshold $\Phi$, as a variant of BoN model. In such cases, both the win-rate and KL divergence can be exactly calculated (Section 6). Finally, they discuss on the conventional evaluation of the win-rate vs KL divergence tradeoff, and they conclude that the previous results are too pessimistic and can be improved with their results in Section 3 and Section 5 (Section 7).

**Claims And Evidence:**

- Claim 1: The conventionally used formula for the KL divergence does not hold and actually gives only an upper bound, especially when N is sufficiently large.
    - Theorem 3.1, 3.4 analyze the tightness of the formula, and Theorem 3.6 shows the looseness when N is sufficiently large.
- Claim 2: The proposed estimator for the KL divergence might give a tighter upper bound than the conventionally used formula.
    - empirically checked in Figure 2-4.
    - However, the claim is theoretically still a conjecture.
- Claim 3: They analyzed the win-rate and show that it can be approximated by $\frac{N}{N+1}$ as expected in previous literature when N is sufficiently small.
    - proved by Theorem 5.3, 5.4.
    - Question: How the equation (34) can be derived?
- Claim 4: In the case of the rewind-and-repeat model $\pi_{\Phi}$ with a threshold $\Phi$, which can be seen as a variant of BoN model, both the KL divergence and win-rate can be explicitly evaluated in terms of the probability distribution of the reference model.
    - provided by Lemma 6.2 and Theorem 6.3
- Claim 5: The previous (empirical) results on the tradeoff curve between the win-rate and KL divergence are too pessimistic and can be improved by their results.
    - theoretically obtained as corollaries of Theorem 3.1, 3.4, 5.3, 5.4. (Theorem A.6, A.7)
    - empirically checked in Figure 7-8.

Overall, their claims are solid and well-supported by theoretical and empirical results.

**Essential References Not Discussed:**

N/A

**Experimental Designs Or Analyses:**

Their experiments seem to be well-designed.

**Methods And Evaluation Criteria:**

The proposed estimator for the KL divergence has some evidences both theoretically and empirically, but the actual guarantee is still a conjecture. However, this is not a fault of the paper since it is explicitly stated in Conjecture 4.4.

**Other Comments Or Suggestions:**

N/A

**Other Strengths And Weaknesses:**

See Claims and Evidences.

**Questions For Authors:**

See Claims and Evidences.

**Relation To Broader Scientific Literature:**

One of the key contributions of this paper is that it fixes the conventional misunderstanding on KL divergence between the BoN and reference models. The other analyses are also valuable to the community of inference-time alignment.

**Theoretical Claims:**

I roughly checked all proofs in Appendix.

---

> ### Author Rebuttal · Authors · 2025-03-31
>
> Thanks for your insightful review and encouraging comments.
>
> > How the equation (34) can be derived?
>
> Eq. (34) is derived by combining Theorem 5.3, i.e., Eq. (30), and Corollary 5.5, i.e., Eq. (33). We will clarify this.
>
> > The proposed estimator for the KL divergence has some evidences both theoretically and empirically, but the actual guarantee is still a conjecture. However, this is not a fault of the paper since it is explicitly stated in Conjecture 4.4.
>
> While we acknowledge this weakness of our work. We would like to mention two points:
> * We have tested this conjecture in tens of thousands of randomly generated numerical examples with varying support sizes and n and have not found a counter example.
> * If we replaced $\epsilon_n$ with $\epsilon_\infty$, then the estimator would lead to an actual upper bound on the KL divergence (Corollary 4.2), which at least suggests that for large $n$ the estimator gives an upper bound.

---

### Decision · Program_Chairs · 2025-05-01

**Decision:**

Accept (poster)

**Comment:**

This paper considers finetuning alignment of generative models via best-of-n policy. In particular, the paper shed insights into the commonly-referred formula $\tilde{KL}_n$ for the regularization term just serves as an upper-bound of the actual value of it and provided a more accurate estimator. The paper also provided a new rejection sampling that outperforms best-of-n theoretically and empirically. The paper is easy to read. All reviewers are positive and the reviews are informative enough to recommend an acceptance.

I recommend the authors to revise the paper based on the reviews and responses. In particular, additional experimental results with real data (conducted/promised to be conducted) during the author response will increase the value of this paper.